# Learning discrete distributions: user vs item-level privacy

**Yuhan Liu**
Cornell University
yl2976@cornell.edu

**Ananda Theertha Suresh**
Google Research
theertha@google.com

**Felix Yu**
Google Research
felixyu@google.com

**Sanjiv Kumar**
Google Research
sanjivk@google.com

**Michael Riley**
Google Research
riley@google.com

## Abstract

Much of the literature on differential privacy focuses on item-level privacy, where loosely speaking, the goal is to provide privacy per item or training example. However, recently many practical applications such as federated learning require preserving privacy for all items of a single user, which is much harder to achieve. Therefore understanding the theoretical limit of user-level privacy becomes crucial.

We study the fundamental problem of learning discrete distributions over $k$ symbols with user-level differential privacy. If each user has $m$ samples, we show that straightforward applications of Laplace or Gaussian mechanisms require the number of users to be $\mathcal{O}(k/(m\alpha^2) + k/\varepsilon\alpha)$ to achieve an $\ell_1$ distance of $\alpha$ between the true and estimated distributions, with the privacy-induced penalty $k/\varepsilon\alpha$ independent of the number of samples per user $m$. Moreover, we show that any mechanism that only operates on the final aggregate counts should require a user complexity of the same order. We then propose a mechanism such that the number of users scales as $\tilde{\mathcal{O}}(k/(m\alpha^2) + k/\sqrt{m}\varepsilon\alpha)$ and hence the privacy penalty is $\tilde{\Theta}(\sqrt{m})$ times smaller compared to the standard mechanisms in certain settings of interest. We further show that the proposed mechanism is nearly-optimal under certain regimes.

We also propose general techniques for obtaining lower bounds on restricted differentially private estimators and a lower bound on the total variation between binomial distributions, both of which might be of independent interest.

## 1 Introduction

### 1.1 Differential privacy

Differential privacy (DP) [Dwork et al., 2006, Dwork and Roth, 2014, Wasserman and Zhou, 2010] has emerged as the standard framework for providing privacy for various statistical problems. Ever since its inception, it has been applied to various statistical and learning scenarios including learning histograms [Dwork et al., 2006, Hay et al., 2010, Suresh, 2019], statistical estimation [Diakonikolas et al., 2015, Kamath et al., 2019, Acharya et al., 2020, Kamath et al., 2020, Acharya et al., 2019a,b], learning machine learning models [Chaudhuri et al., 2011, Bassily et al., 2014, McMahan et al., 2018b, Dwork et al., 2014], hypothesis testing [Aliakbarpour et al., 2018, Acharya et al., 2018], and various other tasks.

Differential privacy is studied in two scenarios, local differential privacy [Kasiviswanathan et al., 2011, Duchi et al., 2013] and global differential privacy [Dwork et al., 2006]. In this paper, we study

the problem under the lens of global differential privacy, where the goal is to protect the privacy of the algorithm outcomes. Before we proceed further, we first define differential privacy.

**Definition 1.** *A randomized mechanism $\mathcal{M} : \mathcal{D} \to \mathcal{R}$ with domain $\mathcal{D}$ and range $\mathcal{R}$ satisfies $(\varepsilon, \delta)$-differential privacy if for any two adjacent datasets $D, D' \in \mathcal{D}$ and for any subset of output $\mathcal{S} \subseteq \mathcal{R}$, it holds that*

$$\Pr[\mathcal{M}(D) \in \mathcal{S}] \leq e^{\varepsilon} \Pr[\mathcal{M}(D') \in \mathcal{S}] + \delta.$$

If $\delta = 0$, then the privacy is also referred to as pure differential privacy.

An important aspect of the above definition is the notion of neighboring or adjacent datasets. If a dataset $D$ is a collection of $n$ items $x_1, x_2, \ldots, x_n$, then typically adjacent datasets are defined as those that differ in a single item $x_i$ [Dwork et al., 2006].

However, in practice, each user may have many items and may wish to preserve privacy for all of them. Hence, this simple definition of item-level neighboring datasets would not be enough. For example, if each user has infinitely many points of the same example, then the bounds become vacuous.

Motivated by this, user-level privacy was proposed recently. Formally, given $s$ users where each user $u$ has $m_u$ items $x_1(u), x_2(u), \ldots x_{m_u}(u)$, then two datasets are adjacent if they differ in data of a single user. For example, in the simple setting when each user has $m$ samples, if two datasets are adjacent in user-level privacy, they could differ in at most $m$ items under the definition of item-level privacy.

Since user-level privacy is more practical, it has been studied in the context of learning machine learning models via federated learning [McMahan et al., 2018b,a, Wang et al., 2019, Augenstein et al., 2019]. The problem of bounding user contributions in user-level privacy in the context of both histogram estimation and learning machine learning models was studied in Amin et al. [2019]. Differentially private SQL with bounded user contributions was proposed in Wilson et al. [2020]. Understanding trade-offs between utility and privacy in the context of user-level global DP is one of the challenges in federated learning [Kairouz et al., 2019, Section 4.3.2]. Kasiviswanathan et al. [2013] studied node differential privacy which guarantees privacy in the event of adding or removing nodes in network data.

Our goal is to understand theoretically the utility-privacy trade-off for user-level privacy and compare it to the item-level counterpart. To this end, we study the problem of learning discrete distributions under user and item-level privacy.

## 1.2 Learning discrete distributions

Learning discrete distributions is a fundamental problem in statistics with practical applications that include language modeling, ecology, and databases. In many applications, the underlying data distribution is private and sensitive e.g., learning a language model from user-typed texts. To this end, learning discrete distributions under differential privacy has been studied extensively with various loss functions and non-asymptotic convergence rates [Braess and Sauer, 2004, Kamath et al., 2015, Han et al., 2015], with local differential privacy [Duchi et al., 2013, Kairouz et al., 2016, Acharya et al., 2019a, Ye and Barg, 2018], with global differential privacy [Diakonikolas et al., 2015, Acharya et al., 2020], and with communication constraints [Barnes et al., 2020, Acharya et al., 2019a], among others.

Before we proceed further, we first describe the learning scenario. Let $p$ be an unknown distribution over symbols $1, 2, \ldots, k$ i.e., $\sum_i p_i = 1$ and $p_i \geq 0$ for all $i \leq k$. Let $\Delta_k$ be the set of all discrete distributions over the domain $[k] := \{1, 2, \ldots, k\}$.

Suppose there are $s$ users indexed by $u$, and let $\mathcal{U}$ denote the set of all users. We assume that each user $u$ has $m$ i.i.d. samples $X^m(u) = [X_1(u), X_2(u), \ldots, X_m(u)] \in \mathcal{X} := [k]^m$ from the same distribution $p$. We extend our results to the case when users have different number of samples in Appendix E. However, we assume that all users have samples from the same distribution throughout the paper. Extending the algorithms to scenarios where users have samples from different distributions is an interesting open direction.

Let $X^s = [(u, X^m(u)) : u \in \mathcal{U}]$ be the set of user and sample pairs. Let $\mathcal{X}^s$ be the collection of all possible user-sample pairs. For an algorithm $A$, let $\hat{p}^A(X^s)$ be its output, a mapping from

$\mathcal{X}^s \mapsto \Delta_k$. The performance for a given sample $X^s$ is measured in terms of $\ell_1$ distance, $\ell_1(p, \hat{p}^A) = \sum_{i=1}^k |p_i - \hat{p}_i^A(X^s)|$. We measure the performance of the estimator for a distribution $p$ by its expectation over the algorithm and samples i.e., $L(A, s, m, p) = \mathbb{E}_{A, X^s}[\ell_1(p, \hat{p}^A(X^s))]$.

We define the user complexity of an algorithm $A$ as the minimum number of users required to achieve error at most $\alpha$ for all distributions:

$$S_{m,\alpha}^A = \min_s \{s : \sup_{p \in \Delta_k} L(A, s, m, p) \le \alpha\}. \tag{1}$$

The min-max user complexity is

$$S_{m,\alpha}^* = \min_A S_{m,\alpha}^A.$$

Well known results on non-private discrete distribution estimation (see [Kamath et al., 2015, Han et al., 2015]) characterize the min-max user complexity as

$$S_{m,\alpha}^* = \Theta\left(\frac{k}{m\alpha^2}\right). \tag{2}$$

Let $\mathcal{A}_{\varepsilon,\delta}$ be the set of all $(\varepsilon, \delta)$ differentially private algorithms. Similar to (1), for a differentially private algorithm $A$, let $S_{m,\alpha,\varepsilon,\delta}^A$ be the minimum of samples necessary to achieve $\alpha$ error for all distributions $p \in \Delta_p$ with $(\varepsilon, \delta)$ differential privacy. We are interested in characterizing and developing polynomial-time algorithms that achieve the min-max user complexity of $(\varepsilon, \delta)$ differentially private mechanisms.

$$S_{m,\alpha,\varepsilon,\delta}^* = \min_{A \in \mathcal{A}_{\varepsilon,\delta}} S_{m,\alpha,\varepsilon,\delta}^A.$$

## 2 Previous results

The min-max rate of learning discrete distributions for item-level privacy, which corresponds to $m = 1$, was studied by Diakonikolas et al. [2015] and Acharya et al. [2020]. They showed that for any $(\varepsilon, \delta)$ estimator,

$$S_{1,\alpha,\varepsilon,\delta}^* = \Theta\left(\frac{k}{\alpha^2} + \frac{k}{\alpha(\varepsilon + \delta)}\right).$$

The goal of our work is to understand the behavior of $S_{m,\alpha,\varepsilon,\delta}^*$ w.r.t. m. We first discuss a few natural algorithms and analyze their user complexities.

One natural algorithm is for each user to sample one item and use known results from item-level privacy. Such a result would yield,

$$S_{m,\alpha,\varepsilon,\delta}^{\text{sample}} = \mathcal{O}\left(\frac{k}{\alpha^2} + \frac{k}{\alpha(\varepsilon + \delta)}\right).$$

The other popular algorithms are Laplace or Gaussian mechanisms that rely on counts of users. For a particular user sample $X^m(u)$, let $N(u) = [N_1(u), \ldots, N_k(u)]$, be the vector of counts. A natural algorithm is to sum all the user contributions to obtain the overall count vector $N$, where the count of a symbol $i$ is given by

$$N_i = \sum_u N_i(u).$$

Finally a non-private estimator can be obtained by computing the empirical estimate:

$$\hat{p}_i^{\text{emp}} = \frac{N_i}{ms}.$$

To obtain a differentially private version of the empirical estimate, one can add Laplace or Gaussian noise with some suitable magnitude. To this end, we need to compute the sensitivity of the empirical estimate.

Recall that two datasets $D, D'$ are adjacent if there exists a single user $u$ such that $N(u, D) \ne N(u, D')$, and $N(v, D) = N(v, D')$ for all $v \in \mathcal{U}$ and $v \ne u$. Therefore the $\ell_1$ sensitivity is

$$\Delta_1(N) = \max_{D, D' \text{ adjacent}} ||N(D) - N(D')||_1 = 2m.$$

and the $\ell_2$ sensitivity is

$$\Delta_2(N) = \max_{D,D' \text{ adjacent}} ||N(D) - N(D')||_2 = \sqrt{2}m.$$

A widely used method is the Laplace mechanism, which ensures $(\varepsilon, 0)$ differential privacy.

**Definition 2.** *Given any function $f$ that maps the dataset to $\mathbb{R}^k$, let the $\ell_1$ sensitivity $\Delta(f) = \max_{D,D' \text{ adjacent}} ||f(D) - f(D')||_1$. The Laplace mechanism is defined as*

$$\mathcal{M}(D, f(\cdot), \varepsilon) = f(D) + (Y_1, \ldots, Y_k),$$

*where $Y_i$ are i.i.d random variables drawn from $Lap(\Delta f/\varepsilon)$.*

The Gaussian mechanism is defined similarly with $\ell_2$ sensitivity and Gaussian noise. We first analyze Laplace and Gaussian mechanisms under user-level privacy.

**Lemma 1.** *For the Laplace mechanism, given by $\hat{p}_i^l = \hat{p}_i^{emp} + \frac{Z_i}{ms}$, where $Z_i = Lap(2m/\varepsilon)$,*

$$S_{m,\alpha,\varepsilon,0}^l = \mathcal{O}\left(\frac{k}{m\alpha^2} + \frac{k}{\alpha\varepsilon}\right).$$

*Similarly if $\varepsilon \leq 1$, for the Gaussian mechanism, given by $\hat{p}_i^g = \hat{p}_i^{emp} + \frac{Z_i}{ms}$, where $Z_i = \mathcal{N}(0, 4\log(1.25/\delta)m^2/\varepsilon^2)$,*

$$S_{m,\alpha,\varepsilon,\delta}^g = \mathcal{O}\left(\frac{k}{m\alpha^2} + \frac{k}{\alpha\varepsilon}\sqrt{\log\frac{1}{\delta}}\right).$$

The proof follows from the definitions of the Laplace and Gaussian mechanisms, which we provide in Appendix A for completeness. The non-private user complexity term $\mathcal{O}(k/(m\alpha^2))$ decreases with the number of samples from user $m$, but somewhat surprisingly the additional term due to privacy $\mathcal{O}(k/\alpha\varepsilon)$ is independent of $m$. In other words, no matter how many samples each user has, it does not help to reduce the privacy penalty in the user complexity. This could be especially troublesome when $m$ gets large, in which case the privacy term dominates the user complexity.

## 3  New results

We first ask if the above results on Laplace and Gaussian mechanisms are tight. We show that they are by proving a lower bound on a wide class of estimators that only rely on the final count. The proof is based on a new coupling technique with details explained in Section 4 .

**Theorem 1.** *Let $\varepsilon + \delta < c$, where $c$ is determined in the proof later. Let $A$ be any $(\varepsilon, \delta)$ mechanism that only operates on summed counts of all users $N = [N_1, N_2, \ldots N_k]$ directly. Then,*

$$S_{m,\alpha,\varepsilon,\delta}^A = \Omega\left(\frac{k}{m\alpha^2} + \frac{k}{\alpha(\varepsilon + \delta)}\right).$$

The above lower bound suggests that any algorithm that only operates on the final count aggregate would incur additional cost for user complexity independent of $m$ due to privacy restriction. However it may not apply to algorithms that do not solely rely on the counts, which justifies the need to design algorithms beyond straightforward applications of the Laplace or Gaussian mechanisms.

We proceed to design algorithms that exceed the above user-complexity limit. The first one is for the dense regime where $k \leq m$: on average each user sees most of the high-probability symbols. The second one is for the sparse regime where $k \geq m$: users don't see many symbols. By combining the two of them, we get the following improved upper bound on min-max user complexity.

**Theorem 2.** *Let $\varepsilon \leq 1$. There exists a polynomial time algorithm $(\varepsilon, \delta)$-differentially private algorithm $A$ such that*

$$S_{m,\alpha,\varepsilon,\delta}^A = \mathcal{O}\left(\log\frac{km}{\alpha} \cdot \max\left(\frac{k}{m\alpha^2} + \frac{k}{\sqrt{m}\alpha\varepsilon}\sqrt{\log\frac{1}{\delta}}, \frac{\sqrt{k}}{\varepsilon}\sqrt{\log\frac{1}{\delta}}\right)\right). \tag{3}$$

The algorithm in Theorem 2 assumes that all users have the same number of samples. When $k$ is large or $\alpha$ is small, the first term in the maximum dominates and we obtain $\tilde{\Theta}(\sqrt{m})$ improvement compared to Laplace and Gaussian mechanisms. In Appendix E, we modify it to the setting when users have different number of samples. The sample complexity is similar to (3), with $m$ replaced by $\bar{m}$, the median of number of samples per user. We also note that our algorithms are designed using high probability arguments, and hence we can easily obtain the sample complexity with logarithmic dependence on the inverse of the confidence parameter.

Finally we provide an information theoretic lower bound for any $(\varepsilon, 0)$-differentially private algorithm:

**Theorem 3.** *Let $\varepsilon \leq 1$. Then*

$$S^*_{m,\alpha,\varepsilon,0} = \Omega\left(\frac{k}{m\alpha^2} + \frac{k}{\sqrt{m}\alpha\varepsilon}\right).$$

Theorems 2 and 3 resolve the user complexity of learning discrete distributions up to log factors and the $\delta$-term in privacy. It would be interesting to see if Theorem 3 can be extended to nonzero values of $\delta$. In the next two sections, we first analyze the lower bounds and then propose algorithms.

## 4 Lower bounds

The $\Omega(k/(m\alpha^2))$ part of the user-complexity lower bounds in Theorem 1 and 3 follows from classic non-private results (2). Therefore in this section we focus on the private part.

### 4.1 Lower bound for restricted estimators

We first start with the lower bound for algorithms that work directly on the counts vector $N = [N_1, N_2, \ldots, N_k]$, even though the learner has access to $\{N(u) : u \in \mathcal{U}\}$. This motivates the definition of restricted estimators, which only depends on some function of the observation rather than the observation itself.

**Definition 3** ($f$-restricted estimators). *Let $f : \mathcal{X}^s \mapsto \mathcal{Y}$ which maps users' data to some domain $\mathcal{Y}$. An estimator $\hat{\theta}$ is $f$-restricted if it has the form $\hat{\theta}(X^s) = \hat{\theta}'(f(X^s))$ for some function $\hat{\theta}'$.*

We generalize Assouad's lemma [Assouad, 1983, Yu, 1997] with differential privacy and the restricted estimators using the recent coupling methods of Acharya et al. [2018, 2020]. These bounds could be of interest in other applications and we describe a general framework where they are applicable.

Let $\mathcal{X}$ be some domain of interest and $\mathcal{P}$ be any set of distributions over $\mathcal{X}$.

Assume that $\mathcal{P}$ is parameterized by $\theta : \mathcal{P} \mapsto \Theta \in \mathbb{R}^d$, i.e. each $p \in \mathcal{P}$ can be uniquely represented by a parameter vector $\theta(p) \in \mathbb{R}^d$. Given $s$ samples from an unknown distribution $p \in \mathcal{P}$, an estimator $\hat{\theta} : \mathcal{X}^s \mapsto \Theta$ takes in a sample from $\mathcal{X}^s$ and outputs an estimation in $\Theta$. Let $\ell : \Theta \times \Theta \mapsto \mathbb{R}_+$ be a pseudo-metric that measures estimation accuracy. For a fixed function $f$, let $\mathcal{A}_f$ be the class of $f$-restricted estimators. We are interested in the min-max risk for $(\varepsilon, \delta)$-DP restricted estimators:

$$L(\mathcal{P}, \ell, \varepsilon, \delta) := \min_{\hat{\theta} \in \mathcal{A}_{\varepsilon,\delta} \cap \mathcal{A}_f} \max_{p \in \mathcal{P}} \mathbb{E}_{X^s \sim p^s}[\ell(\hat{\theta}(X^s), \theta(p))].$$

We need two more definitions to state our results.

**Definition 4** ($f$-identical in distribution). *Given a function $f$, two random variables $X$ and $Y$ are $f$-identical in distribution if $f(X)$ and $f(Y)$ have the same distributions, denoted by $Y \sim_f X$. If $X \sim p$ and $Y \sim p'$, then we can also say $p \sim_f p'$.*

**Definition 5** ($f$-coupling). *Given a function $f$ and two distributions $p, q$, random variables $(X, Y)$ are an $f$-coupling of $p$ and $q$ if $X \sim_f p$ and $Y \sim_f q$. When $f$ is the identity mapping, then an $f$-coupling is same as standard coupling.*

We make the following observation for restricted estimators: since we can only estimate the true parameter $\theta$ through some function $f$ of the observation $X^s$, then any random variable $Y^s$ such that $f(Y^s)$ has the same distribution as $f(X^s)$ would yield the same distribution for restricted estimators $\hat{\theta}$. Thus, if $\hat{\theta}$ could distinguish two distributions $p_1, p_2$ from the space of product distributions

$\mathcal{P}^s := \{p^s : p \in \mathcal{P}\}$, then it should also be able to distinguish $p'_1 \sim_f p_1$ and $p'_2 \sim_f p_2$. We are able to prove tighter lower bounds because $p'_1, p'_2$ (potentially outside of $\mathcal{P}^s$) could be harder to distinguish than the original distributions $p_1, p_2$. This is the most significant difference between our method and [Acharya et al., 2020], whose argument does not capture the above observation for restricted estimators and hence requires designing testing problems within the original class of distributions.

With this intuition, we show a generalization of Assouad's lower bound in Theorem 4. It relies on an extension of the Le Cam's method [Le Cam, 1973, Yu, 1997]. The proofs are in Appendix B.1. For two sequences $X^s$ and $Y^s$, let $d_h(X^s, Y^s) = \sum_{i=1}^s 1_{X_i \neq Y_i}$ denote the Hamming distance.

**Theorem 4** (($\varepsilon, \delta$)-DP Assouad's method for restricted estimators). *Let $\mathcal{V} := \{\pm 1\}^k$ be a hypercube. Consider a set of distributions $\mathcal{P}_\mathcal{V} := \{p_\nu : \nu \in \mathcal{V}\}$ over $\mathcal{X}$. Let for all $u, v \in \mathcal{V}$ the loss $\ell$ satisfies*

$$\ell(\theta(p_u), \theta(p_v)) \geq 2\tau \sum_{i=1}^k 1[u_i \neq v_i]. \tag{4}$$

*For each $i \in [k]$, define the following mixture of product distributions:*

$$p^s_{+i} = \frac{2}{|\mathcal{V}|} \sum_{v \in \mathcal{V}: v_i = +1} p^s_v, \quad p^s_{-i} = \frac{2}{|\mathcal{V}|} \sum_{v \in \mathcal{V}: v_i = -1} p^s_v.$$

*If for all $i \in [k]$ there exists an $f$-coupling $(X^s, Y^s)$ between $p^s_{+i}$ and $p^s_{-i}$ with $\mathbb{E}[d_h(X^s, Y^s)] \leq D$, then for any restricted estimator $\hat{\theta} \in \mathcal{A}_f \cap \mathcal{A}_{\varepsilon, \delta}$,*

$$\sup_{p \in \mathcal{P}} \mathbb{E}_{X^s \sim p^s} \ell(\theta(p), \hat{\theta}(X^s)) \geq \max\left( \frac{\tau}{2} \sum_{i=1}^k (1 - d_{TV}(p^s_{+i}, p^s_{-i})), \frac{k\tau}{2} \left(0.9e^{-10\varepsilon D} - 10D\delta\right) \right).$$

The proof of Theorem 1 follows from Theorem 4. We provide details in Appendix B.2.

*Proof sketch of Theorem 1.* In our problem setting, $\mathcal{X} = [k]^m$ is the domain and $\mathcal{P}$ is the set of multinomial distributions $\mathcal{P} = \{\text{Mul}(m, p) : p \in \Delta_k\}$, where $\text{Mul}(m, p)$ denotes the multinomial distribution. The parameter we are trying to estimate is the underlying $p$ and the loss is $\ell_1$ distance.

We construct $\mathcal{P}_\mathcal{V}$ as follows: let $\alpha \in (0, 1/6)$, and for each $\nu \in \mathcal{V} := \{-1, 1\}^{k/2}$,

$$p_\nu = \text{Mul}\left(m, \frac{1}{k}(1 + 3\alpha\nu_1, 1 - 3\alpha\nu_1, ..., 1 + 3\alpha\nu_{k/2}, 1 - 3\alpha\nu_{k/2})\right). \tag{5}$$

For any $u, v \in \mathcal{V}$, $\ell_1$ distance satisfies (4) with $\tau = 6\alpha/k$.

For restricted estimator $\hat{p}^A$ which only operates on $N = [N_1, ..., N_k]$, for each $i \in [k]$ we can design an $N$-coupling $(X^s, Y^s)$ of $p^s_{+i}$ and $p^s_{-i}$ with $\mathbb{E}[d_h(X^s, Y^s)] \leq 6\alpha s/k + 1 =: D$. Plugging in $\tau$ and $D$ in Theorem 4 yields the desired min-max rate and user complexity. □

### 4.2 Lower bound for the general case

We provide the complete proof of Theorem 3 in Appendix B.3 and sketch an outline here. We use differentially private Fano's method [Acharya et al., 2020, Corollary 4]. We design a set of distributions $\mathcal{P} \subseteq \Delta_k$ such that, $|\mathcal{P}| = \Omega(\exp(k))$, and for each $p, q \in \mathcal{P}$,

$$\ell_1(p, q) = \Omega(\alpha), \ d_{KL}(\text{Mul}(p)||\text{Mul}(q)) = O(m\alpha^2), \ d_{TV}(\text{Mul}(p)||\text{Mul}(q)) = O(\sqrt{m}\alpha^2).$$

Applying Acharya et al. [2020, Corollary 4] with $M = \Omega(\exp(k)), \tau = \alpha, \beta = O(m\alpha^2), \gamma = O(\sqrt{m}\alpha^2)$ yields the result.

## 5 Algorithms

We first propose an algorithm for the dense regime where $k \leq m$. In this regime, on average each user sees most of the high-probability symbols. However, this algorithm does not extend directly to the sparse regime when $k \geq m$. In the sparse regime, we augment the dense algorithm regime with another sub-routine for small probabilities. Both algorithms could be extended to the case when users have different number of samples (see Appendix E).

---
**Algorithm 1** Private hypothesis selection: PHS($\mathcal{H}, D, \alpha, \varepsilon$) [Bun et al., 2019]
---

1: **Input**: $\mathcal{H} = \{H_1, \ldots, H_d\}$ the set of hypotheses, dataset $D$ of $s$ samples drawn i.i.d. from $p \in \mathcal{H}$, accuracy parameter $\alpha \in (0,1)$, privacy parameter $\varepsilon$.

2: **for each** $H_i, H_j \in \mathcal{H}$ **do**

3: $\quad \mathcal{W} = \{x \in \mathcal{X} : H_i(x) > H_j(x)\}$, $p_i = H_i(\mathcal{W})$, $p_j = H_j(\mathcal{W})$.

4: $\quad$ Compute $\hat{\tau} = \frac{1}{s}|\{x \in D : x \in \mathcal{W}\}|$ and

$$\Gamma(H_i, H_j, D) = \begin{cases} s & p_i - p_j \leq 3\alpha; \\ s \cdot \max\{0, \hat{\tau} - (p_j + (3/2)\alpha)\} & \text{otherwise.} \end{cases}$$

5: **end for**

6: For each $H_j \in \mathcal{H}$ compute $S(H_j, D) = \min_{H_k \in \mathcal{H}} \Gamma(H_j, H_k, D)$.

7: **return** random hypothesis $\hat{H}$ such that for each $H_j$:

$$\Pr[\hat{H} = H_j] \propto \exp\left(\frac{S(H_j, D)}{2\varepsilon}\right).$$

---

---
**Algorithm 2** Learning binomial distributions: Binom($D, \varepsilon, \alpha$)
---

1: **Input**: Dataset $D$ of $s$ samples i.i.d. from $\mathrm{Bin}(m, p)$, privacy parameter $\varepsilon$, accuracy parameter $\alpha$.

2: Let $\mathcal{P} = \{0, \frac{c\alpha}{20m}, \frac{2c\alpha}{20m}, \ldots, 1]\}$ and $\mathcal{H} = \{\mathrm{Bin}(m,p) : p \in \mathcal{P}\}$.

3: Run PHS($\mathcal{H}, D, c\alpha/5, \varepsilon$) and obtain $\mathrm{Bin}(m, \hat{p})$.

4: **return** $\hat{p}$.

---

### 5.1 Algorithms for the dense regime

We first motivate our algorithm with an example. Consider a symbol with probability around $1/2$. If $m$ is large, then by the Chernoff bound, such a symbol has counts in the range

$$\left[\frac{m}{2} - \sqrt{\frac{m}{2}\log\frac{2}{\delta}}, \frac{m}{2} + \sqrt{\frac{m}{2}\log\frac{2}{\delta}}\right],$$

with probability $\geq 1 - \delta$. Hence, neighboring datasets differ typically with $\sqrt{m}$ counts. However, in the worst case, they could differ by $m$ and hence standard mechanisms add noise proportional to $m$.

We propose the following alternative method. The count for symbol $i \in [k]$ can take values from $0, 1, \ldots, m$ and is distributed according to $\mathrm{Bin}(m, p_i)$. Thus, we can learn this distribution $\mathrm{Bin}(m, p_i)$ itself to a good accuracy and then estimate $p_i$ from the estimated density of $\mathrm{Bin}(m, p_i)$.

We propose to use the private hypothesis selection algorithm due to Bun et al. [2019] to learn the density of the Binomial distribution. It gives a score for every hypothesis using the Scheffé algorithm [Scheffé, 1947] and then privately selects a hypothesis using the exponential mechanism based on the score functions. For completeness, we state the private hypothesis selection algorithm in Algorithm 1 and its guarantee in Lemma 9 in the Appendix.

Our proposed algorithm for learning Binomial distributions is given in Algorithm 2. We compute a cover of Binomial distributions and use Algorithm 1 to select an estimate of the underlying Binomial distribution. We return the parameter of the Binomial distribution as our estimate for the underlying parameter. Translating the guarantees on total variation distance between binomial distributions to difference of parameters requires a bound on parameter estimation from the binomial density estimation. To this end, we show the following theorem, which might be of independent interest.

**Theorem 5.** *For all $m$ and $p, q$,*

$$\ell_1(\textit{Bin}(m, p), \textit{Bin}(m, q)) = \Theta\left(\min\left(m|p - q|, \frac{\sqrt{m}|p - q|}{\sqrt{p(1-p)}}, 1\right)\right).$$

Due to space constraints, we provide the proof in Appendix C. We show empirically that the bounds in Theorem 5 should hold by estimating the $\ell_1$ distance between $\mathrm{Bin}(m, 0.01)$ and $\mathrm{Bin}(m, 0.011)$.

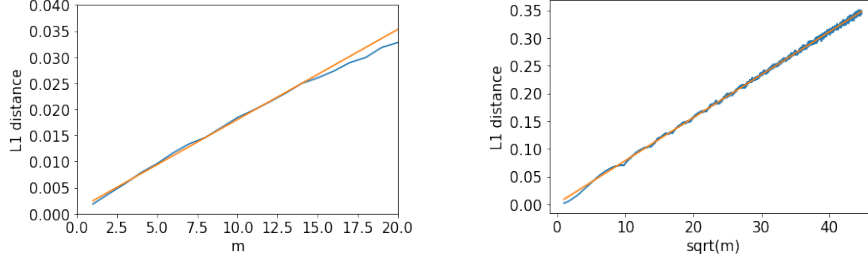

(a) $m \leq 20$. $\ell_1$ distance grows linearly with $m$. (b) Larger values of $m$. $\ell_1$ distance grows as $\sqrt{m}$.

Figure 1: $\ell_1(\mathrm{Bin}(m,p), \mathrm{Bin}(m,q))$ with $p = 0.01$ and $q = 0.011$. We approximate the $\ell_1$ distance by samples. The blue curves are the approximations and orange curves are the best line fit.

---

**Algorithm 3** Dense regime: $\mathrm{Dense}(D, \varepsilon, \delta, \alpha)$

---

1: **Input**: dataset $D$ of $s$ samples i.i.d. from $\mathrm{Mul}(m, p)$, where $p \in \Delta_k$, privacy parameter $\varepsilon, \delta$, accuracy parameter $\alpha$.

2: $\varepsilon' = \frac{\varepsilon}{4\sqrt{k \log(1/\delta)}}, \alpha' = \min\left(\frac{\sqrt{m}\alpha}{2\sqrt{k}}, 1\right)$.

3: For each $i \in [k]$, learn the binomial distribution $\mathrm{Bin}(m, p_i)$ using Algorithm 2, i.e. $\hat{p}_i = \mathrm{Binom}(D_i, \varepsilon', \alpha')$, where $D_i$ is the dataset of counts of symbol $i$ in $D$.

4: **return** $\hat{p} = [\hat{p}_1, \ldots, \hat{p}_k]$.

---

Figure 1 shows that the $\ell_1$ distance grows linearly with $m$ when $m$ is small, and grows linearly with $\sqrt{m}$ when $m$ is large, which illustrates our bounds in Theorem 5.

Combining Lemma 9 with Theorem 5 yields guarantees for Algorithm 2. Its sample complexity and utility are given by Theorem 6. We provide a proof in Appendix D.1.

**Theorem 6.** *Let* $s \geq \frac{16 \log(20m/\alpha\beta)}{\alpha^2} + \frac{16 \log(20m/\alpha\beta)}{\alpha\varepsilon}$. *Given* $s$ *i.i.d. samples from an unknown binomial distribution* $\mathrm{Bin}(m, p)$, *Algorithm* 2 *returns* $\hat{p}$ *such that with probability at least* $1 - \beta$,

$$|p - \hat{p}| \leq \alpha \max\left(\frac{1}{m}, \frac{\sqrt{p(1-p)}}{\sqrt{m}}\right).$$

*Furthermore, Algorithm* 2 *is* $(\varepsilon, 0)$*-differentially private.*

Applying Algorithm 2 independently on each symbol $i$ to learn $p_i$, we obtain Algorithm 3, an $(\varepsilon, \delta)$-private algorithm that learns unknown multinomial distributions under the dense regime. Its user complexity is given by Theorem 7. We provide the proof in Appendix D.2.

**Theorem 7** (Dense regime). *Let* $k \leq m$ *and* $\varepsilon \leq 1$. *Algorithm* 3 *is* $(\varepsilon, \delta)$*-differentially private and has sample complexity given by,*

$$S^A_{m,\alpha,\varepsilon,\delta} = \mathcal{O}\left(\log \frac{km}{\alpha} \cdot \max\left(\frac{k}{m\alpha^2} + \frac{k}{\sqrt{m}\alpha\varepsilon}, \frac{\sqrt{k}}{\varepsilon}\sqrt{\log \frac{1}{\delta}}\right)\right).$$

Theorem 7 has a better dependency on $m$ than that of the Laplace or Gaussian mechanism. Furthermore, even if the number of samples tends to infinity, the number of users is least $\mathcal{O}(\sqrt{k})$.

## 5.2 Algorithms for the sparse regime

We now propose a more involved algorithm for the sparse regime where $m \leq k$. In this regime, users will not see samples from many symbols. A direct application of the private hypothesis selection algorithm would not yield tight bounds in this case.

We overcome this by proposing a new subroutine for estimating symbols with small probabilities, say $p_i \leq 1/m$. In this regime, most symbols appear at most once. Hence, we propose each user sends if a symbol appeared or not i.e., $1_{N_i(u)>0}$. Since $N_i(u)$ is distributed as $\mathrm{Bin}(m, p)$, observe that

$$\mathbb{E}[1_{N_i(u)=0}] = (1 - p_i)^m.$$

---

**Algorithm 4** Estimation of binomial with small $p$: SmallBinom($D, \varepsilon$)

---

1: **Input**: dataset $D$ of $s$ samples i.i.d. from $\text{Bin}(m, p)$, privacy parameter $\varepsilon$.
2: **return** $\hat{p}$ such that:

$$(1 - \hat{p})^m = \max\left(\min\left(\frac{1}{s}\sum_u 1_{N(u)=0} + Z, 1\right), 0\right),$$

where $Z \sim Lap(1/\varepsilon)$.

---

---

**Algorithm 5** Sparse regime: Sparse($D, \varepsilon, \delta, \alpha$)

---

1: Input: dataset $D$ of $s$ i.i.d. samples from $\text{Mul}(m, p), p \in \Delta_k$, privacy parameter $\varepsilon, \delta$, accuracy parameter $\alpha$.
2: $\varepsilon' = \frac{\varepsilon}{8\sqrt{\min(k,m)\log\frac{1}{\delta}}}, \alpha' = \min\left(\frac{\sqrt{m}\alpha}{8\sqrt{k}}, 1\right), \alpha'' = \frac{\alpha}{240}$.
3: $\hat{p} = \text{Dense}(D, \varepsilon', \alpha'')$.
4: Obtain $D_1, \ldots, D_k$ from $D$ where each $D_i$ consists of $s$ i.i.d. samples from $\text{Bin}(m, p_i)$.
5: **for** $i = 1 : k$ **do**
6:   **if** $\hat{p}_i < 2/m$ **then**
7:     $\hat{p}_i \leftarrow \text{SmallBinom}(D_i, \varepsilon')$, where $D_i$ is the dataset of counts of symbol $i$ in $D$.
8:   **end if**
9: **end for**
10: **return** $\hat{p} = [\hat{p}_1, \ldots, \hat{p}_k]$.

---

Hence, if we get a good estimate for this quantity, then since $p_i \leq 1/m$, we can use it to get a good estimate of $p_i$. We describe the details of this approach in Algorithm 4. Its user complexity and utility guarantee are given by Lemma 2, whose proof is in Appendix D.3.

**Lemma 2.** *Let* $p \leq \min(c/m, 1/2)$. *Let the number of users* $s \geq 64e^{3c}\max(c, 1)\log\frac{3}{\beta}$ *and* $s \geq \frac{16e^{3c}}{\alpha^2}\log\frac{3}{\beta} + \frac{16e^{3c}}{\gamma\varepsilon}\log\frac{3}{\beta}$. *Algorithm 4 is* $(\varepsilon, 0)$-*differentially private and returns* $\hat{p}$ *such that with probability at least* $1 - \beta$,

$$|p - \hat{p}| \leq \sqrt{\frac{p\alpha^2}{m}} + \frac{\alpha^2}{m} + \frac{\gamma}{m}.$$

Combining the private hypothesis selection algorithm and the subroutine described in Algorithm 4, we obtain an algorithm for the sparse regime, shown in Algorithm 5. We first estimate $p$ using the private hypothesis selection algorithm. If for some $i$, the estimated probability is too small, we run Algorithm 4 to obtain a more accurate estimate of $p_i$. Theorem 8 gives the user complexity guarantee of Algorithm 5. We provide the proof in Appendix D.4.

**Theorem 8.** *Let* $\varepsilon \leq 1$ *and* $k \geq m$. *Algorithm 5 is* $(\varepsilon, \delta)$-*differentially private algorithm and has sample complexity,*

$$S^A_{m,\alpha,\varepsilon,\delta} = \mathcal{O}\left(\log\frac{km}{\alpha}\cdot\left(\frac{k}{m\alpha^2} + \frac{k}{\sqrt{m}\varepsilon\alpha}\sqrt{\log\frac{1}{\delta}}\right)\right).$$

## 6 Conclusion

We studied user-level differential privacy and its theoretical limit in the context of learning discrete distributions and proposed a near-optimal algorithm. Generalizing the results to non-i.i.d. user data, proposing a more practical algorithm, and characterizing user-level privacy for other statistical estimation problems such as empirical risk minimization are interesting future research directions. Our techniques for obtaining lower bounds on restricted differentially private estimators and the lower bound on the total variation between binomial distributions could be of interest in other scenarios.

# 7 Broader impact

In this work, we propose algorithms that have better privacy-utility trade-offs under global differential privacy compared to those of standard algorithms. Privacy-aware techniques are crucial for widespread use of machine learning leveraging user data. While our work is theoretical in nature, we hope that having higher utility private algorithms would encourage more practitioners to adopt user-level differential privacy in their applications.

# 8 Acknowledgements

Authors thank Jayadev Acharya, Peter Kairouz, and Om Thakkar for helpful comments and discussions.

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
