[Supplementary Material]

# Appendix: Learning discrete distributions: user vs item-level privacy

## A  Proof of Lemma 1

Note that $\hat{p}_i = (N_i + Z_i)/(sm)$. Thus,

$$\mathbb{E}[\ell_1(p, \hat{p})] = \mathbb{E} \sum_{i=1}^{k} |\hat{p}_i - p_i|$$

$$= \sum_{i=1}^{k} \mathbb{E} \left| \frac{N_i + Z_i}{sm} - p_i \right|$$

$$\leq \sum_{i=1}^{k} \mathbb{E} \left| \frac{N_i}{sm} - p_i \right| + \frac{1}{sm} \mathbb{E} \sum_{i=1}^{k} |Z_i|.$$

The first term is upper bounded by $\sqrt{k/(sm)}$ from classic learning bounds for discrete distribution, which can be obtained by applying the Cauchy-Schwartz inequality, and noting that $N_i \sim \text{Bin}(sm, p_i)$,

$$\left( \mathbb{E} \sum_{i=1}^{k} \left| \frac{N_i}{sm} - p_i \right| \right)^2 \leq \mathbb{E} \left[ k \cdot \sum_{i=1}^{k} \left| \frac{N_i}{sm} - p_i \right|^2 \right]$$

$$= k \sum_{i=1}^{k} \mathbb{E} \left[ \left| \frac{N_i}{sm} - p_i \right|^2 \right]$$

$$= k \sum_{i=1}^{k} \frac{Var(N_i)}{(sm)^2} = k \cdot \sum_{i=1}^{k} \frac{p_i(1 - p_i)}{sm}$$

$$\leq k \sum_{i=1}^{k} \frac{p_i}{sm} = \frac{k}{sm}.$$

For Laplace mechanism, $Z_i \sim Lap(2m/\varepsilon)$, we have $\mathbb{E}|Z_i| = 2m/\varepsilon$. Thus,

$$\mathbb{E}[\ell_1(p, \hat{p})] \leq \sqrt{\frac{k}{sm}} + \frac{2k}{s\varepsilon}.$$

For Gaussian mechanism, $Z_i \sim N(0, \sigma^2)$ where $\sigma^2 = 4\log(1.25/\delta)m^2/\varepsilon^2$. Using Jensen's inequality we have $\mathbb{E}|Z_i| \leq \sqrt{\mathbb{E}[Z^2]} = \sigma$. Thus,

$$\mathbb{E}[\ell_1(p, \hat{p})] \leq \sqrt{\frac{k}{sm}} + O\left( \frac{k}{s\varepsilon} \sqrt{\log \frac{1}{\delta}} \right).$$

Setting the right hand side of the above inequalities to be $\leq \alpha$ and rearranging the terms we obtain the desired lower bound on $s$.

## B  Lower bounds

### B.1  Proof of Theorem 4

The proof of Assouad's Lemma relies on Le Cam's method [Le Cam, 1973, Yu, 1997], which provide lower bounds for min-max error in hypothesis testing. Let $\mathcal{P}_1 \subseteq \mathcal{P}$ and $\mathcal{P}_2 \subseteq \mathcal{P}$ be two disjoint subsets of distributions. Let $\hat{\theta} : \mathcal{X}^s \mapsto \{1, 2\}$ be an estimator of the indices, which receives $s$ samples and predicts whether the samples come from $\mathcal{P}_1$ or $\mathcal{P}_2$. We are interested in the worst case error probability

$$P_e(\hat{\theta}, \mathcal{P}_1, \mathcal{P}_2) = \max_{i \in \{1,2\}} \max_{p \in \mathcal{P}_i} \Pr_{X^s \sim p^s} (\hat{\theta}(X^s) \neq i).$$

**Theorem 9** ($(\varepsilon, \delta)$-DP Le Cam's method for restricted tests). *Let $p_1^s \in co(\mathcal{P}_1^s)$ and $p_2^s \in co(\mathcal{P}_2^s)$ where $co(\mathcal{P}_i^s)$ represents the convex hull of $\mathcal{P}_i^s := \{p^s : p \in \mathcal{P}_i\}$. Let $(X^s, Y^s)$ be an $f$-coupling between $p_1^s$ and $p_2^s$ with $\mathbb{E}[d_h(X^s, Y^s)] = D$. Then for $\varepsilon \geq 0, \delta \geq 0$, any $f$-restricted $(\varepsilon, \delta)$-DP hypothesis testing algorithm $\hat{\theta}$ must satisfy*

$$P_e(\hat{\theta}, \mathcal{P}_1, \mathcal{P}_2) \geq \frac{1}{2} \max\{1 - d_{TV}(p_1^s, p_2^s), 0.9e^{-10\varepsilon D} - 10D\delta\}.$$

*Proof.* The first term follows from the classic Le Cam's lower bound (see [Yu, 1997, Lemma 1]). For the second term, let $(X^s, Y^s)$ be an $f$-coupling of $p_1^s, p_2^s$ with $\mathbb{E}[d_h(X^s, Y^s)] \leq D$. Define $\mathcal{W} := \{(x^s, y^s)|d_h(x^s, y^s) \leq 10D\}$ as the set of realizations with Hamming distance at most $10D$. By Markov's inequality,

$$\sum_{(x^s, y^s) \notin \mathcal{W}} \Pr(x^s, y^s) = \Pr(d_h(X^s, Y^s) > 10D) < 0.1 \tag{6}$$

Let $x^s, y^s$ be the realizations of $X^s$ and $Y^s$ respectively and define

$$\Pr(x^s, y^s) := \Pr(X^s = x^s, Y^s = y^s).$$

To avoid confusion, we let $(X')^s$ and $(Y')^s$ be random variables from $p_1^s$ and $p_2^s$ respectively. Let

$$\beta_1 = \Pr_{(X')^s \sim p_1^s}(\hat{\theta}((X')^s) = 2)$$

be the error probability when the underlying data is from distribution $p_1^s$. Similarly define $\beta_2 = \Pr_{(Y')^s \sim p_2^s}(\hat{\theta}((Y')^s) = 1)$. Then

$$\begin{aligned}
\beta_1 &= \Pr_{(X')^s \sim p_1^s}(\hat{\theta}((X')^s) = 2) = \Pr(\hat{\theta}(X^s) = 2) \\
&= \sum_{x^s, y^s} \Pr(X^s = x^s, Y^s = y^s)\Pr(\hat{\theta}(X^s) = p_2|X^s = x^s) \\
&\geq \sum_{x^s, y^s \in \mathcal{W}} \Pr(X^s = x^s, Y^s = y^s)\Pr(\hat{\theta}(X^s) = p_2|X^s = x^s).
\end{aligned}$$

Next we need the group property of differential privacy.

**Lemma 3** (Acharya et al. [2020] Lemma 18). *Let $\hat{\theta}$ be an $(\varepsilon, \delta)$-DP algorithm, then for sequences $x^s, y^s \in \mathcal{X}^s$ such that $d_h(x^s, y^s) \leq t$, we have for all subset $S$ of the output domain,*

$$\Pr(\hat{\theta}(y^s) \in S) \leq e^{t\varepsilon}\Pr(\hat{\theta}(x^s) \in S) + \delta t e^{\varepsilon(t-1)}.$$

Note that

$$1 - \beta_2 = \Pr_{(Y')^s \sim p_2^s}(\hat{\theta}((Y')^s) = 2) = \Pr(\hat{\theta}(Y^s) = 2).$$

By Lemma 3 and (6),

$$\begin{aligned}
1 - \beta_2 &= \sum_{(x^s, y^s) \notin \mathcal{W}} \Pr(x^s, y^s)\Pr(\hat{\theta}(Y^s) = 2|Y^s = y^s) + \sum_{(x^s, y^s) \in \mathcal{W}} \Pr(x^s, y^s)\Pr(\hat{\theta}(Y^s) = 2|Y^s = y^s) \\
&\leq 0.1 + \sum_{(x^s, y^s) \in \mathcal{W}} \Pr(x^s, y^s)(e^{10\varepsilon D}\Pr(\hat{\theta}(X^s) = 2|X^s = x^s) + 10D\delta e^{\varepsilon(10D-1)}) \\
&\leq 0.1 + \beta_1 e^{10\varepsilon D} + 10D\delta e^{10\varepsilon D}.
\end{aligned}$$

Similarly we have

$$1 - \beta_1 \leq 0.1 + \beta_2 e^{10\varepsilon D} + 10D\delta e^{10\varepsilon D}.$$

Adding the two inequalities and rearranging the terms we obtain

$$\beta_1 + \beta_2 \geq \frac{1.8 - 10D\delta e^{10\varepsilon D}}{1 + e^{10\varepsilon D}} \geq 0.9e^{-10\varepsilon D} - 10D\delta,$$

which yields the desired lower bound.

$\square$

We now have the necessary ingredients for the Assouad's lower bound. The final step is to apply the classic Assouad's Lemma [Yu, 1997]:

**Theorem 10** (Assouad's Lemma). *Consider a set of distributions $\mathcal{P}_\mathcal{V}$ indexed by the hypercube $\mathcal{V} := \{\pm 1\}^k$. Using the same definitions as in Theorem 4, $\forall i \in [k]$, let $\phi_i : \mathcal{X}^s \mapsto \{-1, 1\}$ be test for $p^s_{+i}$ and $p^s_{-i}$. Then for any estimator $\hat{\theta}$*

$$\sup_{p \in \mathcal{P}} \mathbb{E}_{X^s \sim p^s} \ell(\theta(p), \hat{\theta}(X^s)) \geq \frac{\tau}{2} \sum_{i=1}^{k} \inf_{\phi_i} ( \Pr_{X^s \sim p^s_{+i}} (\phi_i(X^s) \neq 1) + \Pr_{X^s \sim p^s_{-i}} (\phi_i(X^s) \neq -1)). \quad (7)$$

Note that the summand in (7) is the error probability of hypothesis testing between the mixtures $p^s_{+i}$ and $p^s_{-i}$. Applying Theorem 9 completes the proof.

## B.2 Detailed proof of Theorem 1

*Proof.* Let $\mathcal{P}_\mathcal{V}$ be given by (5). For $p_v \in \mathcal{P}_\mathcal{V}$, let $q_v = \theta(p_v)$ be the underlying discrete distribution over $k$ symbols. Then for $u, v \in \mathcal{V}$,

$$\ell_1(\theta(p_u), \theta(p_v)) = \ell_1(q_u, q_v) = \frac{12\alpha}{k} \sum_{i=1}^{k/2} 1[u_i \neq v_i],$$

as one different coordinate between $q_u$ and $q_v$ leads to $l_1$ distance of $12\alpha/k$. Therefore $\tau = 6\alpha/k$. Define the mixtures as

$$p^s_{+i} = \frac{2}{|\mathcal{V}|} \sum_{v \in \mathcal{V}: v_i = +1} p^s_v, \quad p^s_{-i} = \frac{2}{|\mathcal{V}|} \sum_{v \in \mathcal{V}: v_i = -1} p^s_v.$$

It is helpful to look at the underlying distribution of all samples from users.

$$q^{sm}_{+i} = \frac{2}{|\mathcal{V}|} \sum_{v \in \mathcal{V}: v_i = +1} q^{sm}_v, \quad q^{sm}_{-i} = \frac{2}{|\mathcal{V}|} \sum_{v \in \mathcal{V}: v_i = -1} q^{sm}_v.$$

Note that $p^s_{\pm i}, q^s_{\pm i}$ are not necessarily product distributions.

By [Acharya et al., 2020, Lemma 14], there exists a coupling $(U^{sm}, V^{sm})$ between $q^{sm}_{+i}$ and $q^{sm}_{-i}$ such that $\mathbb{E}[d_h(U^{sm}, V^{sm})] \leq 6\alpha sm/k$ (each $U_i, V_i \in [k]$). We construct $X^s = [X_1, ..., X_s]$ and $Y^s = [Y_1, ..., Y_s]$ using this coupling (each $X_i, Y_i \in \mathbb{R}^k$ is the count of symbol $i \in [k]$).

For each realization of $U^{sm}, V^{sm}$, suppose there are $l$ different coordinates, i.e. $d_h(U^{sm}, V^{sm}) = l$, we move all different coordinates to the front so that only the first $\lceil l/m \rceil \leq l/m + 1$ users would have different data. Name the rearranged sequence as $(U')^{sm}, (V')^{sm}$. Then we let user $u$ get data from the $m(u-1) + 1$ to $mu$ coordinates of $(U')^{sm}$ and $(V')^{sm}$ respectively and compute the counts of each symbol to obtain $X^s, Y^s$. Therefore,

$$\mathbb{E}[d_h(X^s, Y^s)] \leq \frac{1}{m} \mathbb{E}[d_h(U^{sm}, V^{sm})] + 1 \leq \frac{6s\alpha}{k} + 1.$$

Rearranging the coordinates of $U^{sm}, V^{sm}$ would not change the total count $N$, and hence $(X^s, Y^s)$ is an $N$-coupling. As a result.

$$\sup_{p \in \mathcal{P}} \mathbb{E}[\ell_1(p, \hat{p})] \geq 3\alpha(0.9e^{-10\varepsilon(6s\alpha/k+1)} - 10\delta(6s\alpha/k + 1)).$$

Choosing $\alpha = \min\{\frac{0.1k}{60s(\varepsilon+\delta)}, \frac{1}{3}\}$ yields,

$$\sup_{p \in \mathcal{P}} \mathbb{E}[\ell_1(p, \hat{p})] \geq \min\left\{\frac{k}{200s(\varepsilon + \delta)}, 1\right\} \left(0.9 \exp\left\{-\frac{0.1\varepsilon}{\varepsilon + \delta} - 10\varepsilon\right\} - \frac{0.1\delta}{\varepsilon + \delta} - 10\delta\right).$$

When $\varepsilon + \delta \leq 0.07$,

$$\sup_{p \in \mathcal{P}} \mathbb{E}[\ell_1(p, \hat{p})] \geq \min\left\{\frac{k}{200s(\varepsilon + \delta)}, 1\right\} \left(0.9\left(1 - \frac{0.1\varepsilon}{\varepsilon + \delta} - 10\varepsilon\right) - \frac{0.1\delta}{\varepsilon + \delta} - 10\delta\right)$$

$$\geq \min\left\{\frac{k}{200s(\varepsilon + \delta)}, 1\right\} (0.9 - 0.1 - 10(\varepsilon + \delta))$$

$$\geq 0.1 \min\left\{\frac{k}{200s(\varepsilon + \delta)}, 1\right\}.$$

Setting the left hand side to be at most $\alpha$ and rearranging the terms, we obtain the desired lower bound for $s$. $\qquad\square$

### B.3 Fano's Lower bound for restricted differentially-private estimators

In this section we provide learning lower bound for restricted estimators under pure differential privacy using Fano's method. First we provide a theorem for restricted estimators like the one we proposed for Assouad's, which might be of general interest.

**Theorem 11** ($\varepsilon$-DP Fano's lower bound for restricted estimators)**.** *Given a family of distributions $\mathcal{P}$ over $\mathcal{X}$ parameterized by $\theta : \mathcal{P} \mapsto \Theta$, and let $\hat{\theta}$ be an $f$-restricted estimator. Let $\mathcal{V} = \{p_1, ..., p_M\} \subseteq \mathcal{P}$ such that for all $i \neq j$,*

1. *$\ell(\theta(p_i), \theta(p_j)) \geq \alpha$*

2. *$d_{KL}(p_i^s, p_j^s) \leq \beta$*

3. *there exists an $f$-coupling $(X^s, Y^s)$ of $p_i^s, p_j^s$ such that $\mathbb{E}[d_h(X^s, Y^s) \leq D]$*

*then*

$$L(\mathcal{P}, l, \varepsilon, 0) := \inf_{\hat{\theta}} \sup_{p \in \mathcal{P}} \mathbb{E}_{X^s \sim p^s} \left[ \ell(\hat{\theta}(X^s), \theta(p)) \right]$$

$$\geq \max \left\{ \frac{\alpha}{2} \left( 1 - \frac{\beta + \log 2}{\log M} \right), 0.4\alpha \min \left\{ 1, \frac{M}{e^{10\varepsilon D}} \right\} \right\}. \tag{8}$$

*Proof.* The first term of (8) follows from the non-private Fano's inequality. We now prove the second term. For an observation $X^s \in \mathcal{X}^s$

$$\hat{p}(X^s) := \arg\min_{p \in \mathcal{V}} \ell(\theta(p), \hat{\theta}(X^s))$$

is the distribution in $\mathcal{P}$ closest to the output of our estimator. Since we require that $\hat{\theta}$ to be $\varepsilon$-DP, $\hat{p}$ is also $\varepsilon$-DP. By triangle inequality, for all $p \in \mathcal{P}$

$$\ell(\theta(\hat{p}), \theta(p)) \leq \ell(\theta(\hat{p}), \hat{\theta}(X^s)) + \ell(\theta(p), \hat{\theta}(X^s)) \leq 2\ell(\theta(p), \hat{\theta}(X^s)).$$

Thus

$$\sup_{p \in \mathcal{P}} \mathbb{E}_{X^s \sim p^s} \left[ \ell(\hat{\theta}(X^s), \theta(p)) \right] \geq \max_{p \in \mathcal{V}} \mathbb{E}_{X^s \sim p^s} \left[ \ell(\hat{\theta}(X^s), \theta(p)) \right]$$

$$\geq \frac{1}{2} \max_{p \in \mathcal{V}} \mathbb{E}_{X \sim p} \left[ \ell(\theta(\hat{p}), \theta(p)) \right]$$

$$\geq \frac{\alpha}{2} \max_{p \in \mathcal{V}} \Pr_{X \sim p} (\hat{p}(X^s) \neq p)$$

$$\geq \frac{\alpha}{2M} \sum_{p \in \mathcal{V}} \Pr_{X \sim p} (\hat{p}(X^s) \neq p). \tag{9}$$

Let $\beta_i = \Pr_{X^s \sim p_i^s}(\hat{p}(X^s) \neq p_i)$. For a fixed $j \neq i$, let $(X^s, Y^s)$ be the $f$-coupling of $p_i^s, p_j^s$ in condition 3. By definition, for $(X')^s \sim p_i^s$, we have $(X')^s \sim_f X^s$ so that $\hat{p}((X')^s)$ and $\hat{p}(X^s)$ have the same distributions, i.e. for all $p \in \mathcal{V}$,

$$\Pr_{(X')^s \sim p_i^s} (\hat{p}((X')^s) = p) = \Pr(\hat{p}(X^s) = p).$$

Same holds for $\hat{p}(Y^s)$ and $\hat{p}((Y')^s)$ such that $(Y')^s \sim p_j^s$.

By Markov's inequality,
$$\Pr(d_h(X^s, Y^s) > 10D) < 1/10.$$
Let $\mathcal{W} := \{(x^s, y^s) | d_h(x^s, y^s) \leq 10D\}$ and $\Pr(x^s, y^s) := \Pr(X^s = x^s, Y^s = y^s)$. Then

$$1 - \beta_j = \Pr_{(Y')^s \sim p_j^s} (\hat{p}((Y')^s) = p_j) = \Pr(\hat{p}(Y^s) = p_j)$$

$$\leq \sum_{(x^s, y^s) \in \mathcal{W}} \Pr(x^s, y^s) \Pr(\hat{p}(Y^s) = p_j | Y^s = y^s) + \sum_{(x^s, y^s) \notin \mathcal{W}} \Pr(x^s, y^s) \cdot 1.$$

Therefore

$$\sum_{(x^s, y^s) \in \mathcal{W}} \Pr(x^s, y^s) \Pr(\hat{p}(Y^s) = p_j | Y^s = y^s) \geq 0.9 - \beta_j.$$

Furthermore

$$\Pr_{(X')^s \sim p_i^s} (\hat{p}((X')^s) = p_j) = \Pr(\hat{p}(X^s) = p_j)$$

$$\geq \sum_{(x^s, y^s) \in \mathcal{W}} \Pr(x^s, y^s) \Pr(\hat{p}(X^s) = p_j | X^s = x^s)$$

$$\geq \sum_{(x^s, y^s) \in \mathcal{W}} \Pr(x^s, y^s) e^{-10\varepsilon D} \Pr(\hat{p}(Y^s) = p_j | Y^s = y^s)$$

$$\geq (0.9 - \beta_j) e^{-10\varepsilon D},$$

where the second inequality is due to $\hat{p}$ is $\varepsilon$-DP and $d_h(x^s, y^s) \leq 10D$. The above inequality holds for all $j \neq i$. Thus summing over all $j \neq i$ we obtain

$$\beta_i = \sum_{j \neq i} \Pr_{X^s \sim p_j^s} (\hat{p}(X^s) = p_j) \geq \left( 0.9(M-1) - \sum_{j \neq i} \beta_j \right) e^{-10\varepsilon D}.$$

Summing over all $i \in \{1, ..., M\}$

$$\sum_{i=1}^{M} \beta_i \geq \left( 0.9M(M-1) - (M-1) \sum_{i=1}^{M} \beta_i \right) e^{-10\varepsilon D}.$$

Rearranging the terms

$$\sum_{i=1}^{M} \beta_i \geq \frac{0.9M(M-1)}{M-1+e^{10\varepsilon D}} \geq 0.8M \min \left\{ 1, \frac{M}{e^{10\varepsilon D}} \right\}.$$

Combining with (9) gives the desired lower bound. $\qquad \square$

*Proof of Theorem 3.* We apply Theorem with $f$ as the identity mapping. In this case it is the same as [Acharya et al., 2020, Theorem 2].

Assume $k$ is even. From Yu [1997], there exists $\mathcal{V} \subseteq \{-1, 1\}^{k/2}$ and a universal $c_0 > 0$ such that $|\mathcal{V}| \geq \exp(c_0 k/2)$, each pair at least $k/6$ apart in Hamming distance. Given $\alpha \in (0, 1/6)$, define a family of multinomial distributions $\mathcal{P}_\nu$ which consists of the following distributions indexed by $\nu = (\nu_1, ..., \nu_{k/2}) \in \mathcal{V}$,

$$p_\nu = \text{Mul} \left( m, \frac{1}{k} (1 + 3\alpha\nu_1, 1 - 3\alpha\nu_1, ..., 1 + 3\alpha\nu_{k/2}, 1 - 3\alpha\nu_{k/2}) \right).$$

For $v \in \mathcal{V}$, let $q_v = \theta(p_v)$ be the underlying $k$-ary distribution. Thus for each pair of distributions $p_u, p_v$ from this family we have $\ell_1(\theta(p_u), \theta(p_v)) = \ell_1(q_u, q_v) \geq 12\alpha/k \cdot k/6 = 2\alpha$. Furthermore,

$$d_{KL}(q_u || q_v) \leq \chi^2(q_u || q_v) = \sum_{x=1}^{k} \frac{(q_u(x) - q_v(x))^2}{q_v(x)} \leq 100\alpha^2,$$

$$d_{KL}(p_u || p_v) = m d_{KL}(q_u || q_v) \leq 100m\alpha^2,$$
$$d_{KL}(p_u^s || p_v^s) = s d_{KL}(p_u || p_v) \leq 100sm\alpha^2.$$

Since $f$ is set to be the identity, we just need to design a coupling with appropriate Hamming distance for each pair $p_u^s, p_v^s, u, v \in \nu$. To this end we need the following lemma from den Hollander [2012].

**Lemma 4** (Maximal coupling, den Hollander [2012])**.** *Given distributions $q_1, q_2$ over some domain $\mathcal{X}$, there exists a coupling $(X^s, Y^s)$ between $q_1^s$ and $q_2^s$ such that*

$$\mathbb{E}[d_h(X^s, Y^s)] = s \cdot d_{TV}(q_1, q_2).$$

From Lemma 4 there exists a coupling $(X^s, Y^s)$ between $p_u^s$ and $p_v^s$ such that
$$\mathbb{E}[d_h(X^s, Y^s)] = s \cdot d_{TV}(p_u, p_v).$$
Using Pinsker's inequality, we have
$$d_{TV}(p_u, p_v) \leq \sqrt{\frac{1}{2} d_{KL}(p_u||p_v)} \leq 10\sqrt{m}\alpha.$$
Therefore $\mathbb{E}[d_h(X^s, Y^s)] \leq 10s\sqrt{m}\alpha$. Applying Lemma 11 yields,
$$\sup_{p \in \mathcal{P}} \mathbb{E}[\ell_1(\hat{p}, p)] \geq \max \left\{ \alpha \left( 1 - \frac{100sm\alpha^2 + \log 2}{c_0 k/2} \right), 0.8\alpha \min \left\{ 1, \frac{e^{c_0 k/2}}{e^{100\varepsilon s\sqrt{m}\alpha}} \right\} \right\}.$$
Note that this holds for all $\alpha$. Choose $\alpha = \min\{\frac{1}{6}, \sqrt{\frac{k}{sm}}\}$ and $\alpha = \min\{\frac{1}{6}, \frac{c_0 k}{200s\sqrt{m}\varepsilon}\}$ respectively we get
$$\sup_{p \in \mathcal{P}} \mathbb{E}[\ell_1(\hat{p}, p)] \geq \max \left\{ C_1 \sqrt{\frac{k}{sm}}, C_2 \frac{k}{s\varepsilon} \right\} = \Omega \left( \sqrt{\frac{k}{sm}} + \frac{k}{s\sqrt{m}\varepsilon} \right).$$
Given desired accuracy $\alpha$, setting $\sup_{p \in \mathcal{P}} \mathbb{E}[\ell_1(\hat{p}, p)] \leq \alpha$ gives the desired user complexity bound. $\qquad \square$

## C  Bounds on total variation between binomial distributions

We divide the proof of Theorem 5 into two parts. We prove the upper bound in Lemma 5 and the lower bound in Lemma 8.

We first prove an upper bound on the total variation distance between binomial distributions in terms of the parameters.

**Lemma 5.** *There is a constant $b$ such that for all $m$ and $p, q$,*
$$\ell_1(Bin(m, p), Bin(m, q)) \leq 2\min \left( m|p - q|, \frac{\sqrt{m}|p - q|}{\sqrt{p(1 - p)}}, 1 \right).$$

*Proof.* First observe that by definition,
$$\ell_1(\mathrm{Bin}(m, p), \mathrm{Bin}(m, q)) \leq 2. \qquad (10)$$
Secondly, since $\ell_1$ distance of product distributions is at most the sum of $\ell_1$ distances,
$$\ell_1(\mathrm{Bin}(m, p), \mathrm{Bin}(m, q)) \leq m \cdot \ell_1(\mathrm{Ber}(p), \mathrm{Ber}(q)) \leq 2m|p - q|. \qquad (11)$$
Finally, by Pinsker inequality and the fact that KL divergence of product distributions is the sum of individual KL divergences,
$$\ell_1(\mathrm{Bin}(m, p), \mathrm{Bin}(m, q)) \leq \sqrt{\frac{1}{2} \cdot D(\mathrm{Bin}(m, q)||\mathrm{Bin}(m, p))}$$
$$= \sqrt{\frac{m}{2} \cdot D(\mathrm{Ber}(q)||\mathrm{Ber}(p))}$$
$$\leq \sqrt{\frac{m(p - q)^2}{2p(1 - p)}}, \qquad (12)$$
where the last inequality follows by observing that
$$D(\mathrm{Ber}(q)||\mathrm{Ber}(p)) = q\log\frac{q}{p} + (1 - q)\log\frac{1 - q}{1 - p}$$
$$= q\log\left(1 + \frac{q - p}{p}\right) + (1 - q)\log\left(1 + \frac{p - q}{1 - p}\right)$$
$$\leq q \cdot \frac{q - p}{p} + (1 - q) \cdot \frac{p - q}{1 - p}$$
$$= \frac{(q - p)^2}{p(1 - p)}. \qquad (13)$$
Combining (10), (11), and (12) yields the lemma. $\qquad \square$

**Lemma 6.** *Let c be a constant. If $mp < c$ and $p \leq 1/2$, then*

$$\ell_1(Bin(m,p), Bin(m,q)) \geq \frac{e^{-\frac{3c}{2}}}{2} \min(m|p-q|, 1).$$

*Proof.* By definition,

$$\ell_1(\mathrm{Bin}(m,p), \mathrm{Bin}(m,q)) \geq |(1-p)^m - (1-q)^m|.$$

We first consider the case $q \geq p$. Simplifying the above bound,

$$
\begin{aligned}
(1-p)^m - (1-q)^m &= (1-p)^m \left(1 - \frac{(1-q)^m}{(1-p)^m}\right) \\
&= (1-p)^m \left(1 - \left(1 - \frac{q-p}{1-p}\right)^m\right) \\
&\overset{(a)}{\geq} (1-p)^m \left(1 - e^{\frac{-m(q-p)}{1-p}}\right) \\
&\overset{(b)}{\geq} (1-p)^m \left(1 - e^{-2m(q-p)}\right) \\
&\geq (1-p)^m \left(1 - e^{-1.5m(q-p)}\right) \\
&\geq (1-p)^m \left(1 - e^{-1.5 \min(m(q-p), 0.5)}\right) \\
&\overset{(c)}{\geq} (1-p)^m \min(m(q-p), 0.5) \\
&\overset{(d)}{\geq} e^{-1.5mp} \min(m(q-p), 0.5) \\
&\overset{(e)}{\geq} e^{-1.5c} \min(m(q-p), 0.5).
\end{aligned}
$$

$(a)$ follows by $1 - x \leq e^{-x}$ and $(b)$ follows as $p \leq 1/2$. $(c)$ and $(d)$ follows as $e^{-1.5x} \leq 1 - x$ for $x \leq 1/2$. $(e)$ follows by the bound on $p$. For $q \leq p$,

$$
\begin{aligned}
(1-q)^m - (1-p)^m &= (1-p)^m \left(\frac{(1-q)^m}{(1-p)^m} - 1\right) \\
&= (1-p)^m \left(\left(1 + \frac{p-q}{1-p}\right)^m - 1\right) \\
&\geq (1-p)^m \left((1+p-q)^m - 1\right) \\
&\overset{(a)}{\geq} (1-p)^m m(p-q) \\
&\geq e^{-1.5mp} m(p-q) \\
&\geq e^{-1.5c} m(p-q),
\end{aligned}
$$

$(a)$ follows from the Bernoulli inequality: $(1+x)^n \geq 1 + nx$ for $x \geq -1$. The last inequalities are similar to the last two inequalities for $q \leq p$ case. Combining the above two results, we get

$$|(1-q)^m - (1-p)^m| \geq e^{-1.5c} \min(m|q-p|, 0.5). \tag{14}$$

$\square$

**Lemma 7.** *Let $c > 2$, $m \geq 3$, and $p \leq 1/2$. If $mp \geq c$, then*

$$\ell_1(Bin(m,p), Bin(m,q)) \geq \frac{1}{350} \min\left(\frac{\sqrt{m}|p-q|}{\sqrt{p(1-p)}}, 1\right).$$

*Proof.* Let $q' = p + \sqrt{\frac{p}{8m}}$ if $q > p + \sqrt{\frac{p}{8m}}$, $q' = p - \sqrt{\frac{p}{8m}}$ if $q \leq p - \sqrt{\frac{p}{8m}}$, else $q' = q$. Since $q'$ lies in between $p$ and $q$,

$$\ell_1(\mathrm{Bin}(m,p), \mathrm{Bin}(m,q)) \geq \ell_1(\mathrm{Bin}(m,p), \mathrm{Bin}(m,q')).$$

Furthermore, observe that

$$\frac{3}{4} \leq 1 - \frac{1}{\sqrt{8c}} \leq 1 - \sqrt{\frac{1}{8pm}} \leq \frac{q'}{p} \leq 1 + \sqrt{\frac{1}{8pm}} \leq 1 + \frac{1}{\sqrt{8c}} \leq \frac{5}{4}.$$

By [Adell and Jodrá, 2006, Proposition 2.3], for any two binomial distributions,

$$\ell_1(\mathrm{Bin}(m,p), \mathrm{Bin}(m,q')) = m \int_{u=\min(p,q')}^{\max(p,q')} \Pr(\mathrm{Bin}(m-1,u) = k-1) du,$$

, where $\lceil m \min(p,q') \rceil \leq k \leq \lceil m \max(p,q') \rceil$. Furthermore, observe that

$$\lceil m \min(p,q') \rceil \geq \lceil m \min(mp, 3mp/4) \rceil \geq \lceil 3/2 \rceil \geq 2.$$

Similarly,

$$m - k \geq m - \lceil m \max(p,q') \rceil \geq m - \lceil 5mp/4 \rceil \geq m - 1 - 5mp/4 \geq m - 1 - 5m/8 \geq 3m/8 - 1 \geq 1/8.$$

Since $m-k$ is an integer, $m - k \geq 1$. In order to bound the above quantity further, we first lower bound Binomial coefficients.

$$\Pr(\mathrm{Bin}(m,p) = k) = \binom{m}{k} p^k (1-p)^{m-k}.$$

Recall that by Sterling's approximation, for all $m \geq 1$,

$$\sqrt{2\pi} m^{m+0.5} e^{-m} \leq m! \leq e m^{m+0.5} e^{-m}.$$

Hence for $1 \leq k \leq m-1$,

$$\binom{m}{k} = \frac{m!}{k!(m-k)!}$$

$$\geq \frac{\sqrt{2\pi}}{e^2} \frac{m^{m+0.5} e^{-m}}{k^{k+0.5} e^{-k} (m-k)^{m-k+0.5} e^{-m+k}}$$

$$= \frac{\sqrt{2\pi}}{e^2 \sqrt{m}} \cdot \frac{1}{\sqrt{k/m}\sqrt{1-k/m}} \cdot \frac{1}{(k/m)^k (1-k/m)^{m-k}}.$$

Hence,

$$\Pr(\mathrm{Bin}(m,p) = k) \geq \frac{\sqrt{2\pi}}{e^2 \sqrt{m}} \cdot \frac{1}{\sqrt{k/m}\sqrt{1-k/m}} \cdot \frac{p^k (1-p)^{m-k}}{(k/m)^k (1-k/m)^{m-k}}$$

$$= \frac{\sqrt{2\pi}}{e^2 \sqrt{m}} \cdot \frac{1}{\sqrt{k/m}\sqrt{1-k/m}} \cdot e^{-mD(k/m\|p)}$$

$$\geq \frac{\sqrt{2\pi}}{e^2 \sqrt{m}} \cdot \frac{1}{\sqrt{k/m}\sqrt{1-k/m}} \cdot e^{-m\frac{(k/m-p)^2}{p(1-p)}}$$

$$\geq \frac{\sqrt{2\pi}}{e^2} \cdot \frac{1}{\sqrt{k}} \cdot e^{-m\frac{(k/m-p)^2}{p(1-p)}}.$$

The second inequality follows from (13). Hence for $\lceil m \min(p,q') \rceil \leq k \leq \lceil m \max(p,q') \rceil$,

$$\Pr(\mathrm{Bin}(m,u) = k-1) \geq \frac{\sqrt{2\pi}}{e^2} \cdot \frac{1}{\sqrt{k-1}} \cdot e^{-m\frac{((k-1)/(m-1)-u)^2}{u(1-u)}}$$

$$\overset{(a)}{\geq} \frac{2\sqrt{2\pi}}{5e^2} \cdot \frac{1}{\sqrt{mp}} \cdot e^{-m\frac{((k-1)/(m-1)-u)^2}{u(1-u)}}$$

$$\geq \frac{2\sqrt{\pi}}{5e^2} \cdot \frac{1}{\sqrt{mp(1-p)}} \cdot e^{-m\frac{((k-1)/(m-1)-u)^2}{u(1-u)}},$$

where $(a)$ follows by observing that $k - 1 \leq \lceil m \max(p, q') \rceil - 1 \leq m \max(p, q') \leq 5mp/4$. Furthermore, since $3p/4 \leq q' \leq 5p/4$ and the minimum of $u(1 - u)$ occurs in the extremes,

$$\min_{\min(p,q') \leq u \leq \max(p,q')} u(1 - u) \geq \min_{3p/4 \leq u \leq 5p/4} u(1 - u)$$

$$\geq \min\left(\frac{(1 - 3p/4)3p}{4}, \frac{(1 - 5p/4)5p}{4}\right)$$

$$\geq \frac{15p}{32}.$$

We now bound $((k - 1)/(m - 1) - u)^2$.

$$\max_u \frac{k - 1}{m - 1} - u \leq \frac{k}{m} - u \leq \max(p, q') + \frac{1}{m} - \min(p, q').$$

Similarly,

$$\min_u \frac{k - 1}{m - 1} - u \geq \frac{k - 1}{m - 1} - \min(p, q')$$

$$= \frac{k}{m} + \frac{m - k}{m(m - 1)} - \max(p, q')$$

$$\geq \frac{k}{m} + \frac{1}{m} - \max(p, q')$$

$$\geq \min(p, q') + \frac{1}{m} - \max(p, q').$$

Hence, since $(a + b)^2 \leq 2a^2 + 2b^2$,

$$\max_u \left(\frac{k}{m} - u\right)^2 \leq 2\left(\max(p, q') - \min(p, q')\right)^2 + \frac{2}{m^2}.$$

Hence,

$$e^{-m\frac{((k-1)/(m-1)-u)^2}{u(1-u)}} \geq e^{-\frac{8m}{p}\left(\frac{1}{m^2} + (p - q')^2\right)} \geq e^{-\frac{64m}{15p}\left(\frac{1}{m^2} + \frac{p}{8m}\right)} \geq e^{-\frac{32}{15} - \frac{8}{15}} \geq e^{-8/3}.$$

Combining the results, we get

$$\ell_1(\mathrm{Bin}(m, p), \mathrm{Bin}(m, q')) = m \int_{u=\min(p,q')}^{\max(p,q')} \Pr(\mathrm{Bin}(m - 1, u) = k - 1)du$$

$$\geq \frac{m\sqrt{\pi}e^{-8/3}}{2e^2} \int_{u=\min(p,q')}^{\max(p,q')} \frac{m}{\sqrt{mp(1 - p)}}$$

$$\geq \frac{\sqrt{\pi}e^{-8/3}}{2e^2} \frac{\sqrt{m}|p - q'|}{\sqrt{p(1 - p)}}$$

$$\geq \frac{\sqrt{\pi}e^{-8/3}}{2e^2} \min\left(\frac{\sqrt{m}|p - q|}{\sqrt{p(1 - p)}}, \frac{1}{\sqrt{8}}\right)$$

$$\geq \frac{\sqrt{\pi}e^{-8/3}}{2\sqrt{8}e^2} \min\left(\frac{\sqrt{m}|p - q|}{\sqrt{p(1 - p)}}, 1\right)$$

$$\geq \frac{1}{350} \min\left(\frac{\sqrt{m}|p - q|}{\sqrt{p(1 - p)}}, 1\right).$$

$\square$

**Lemma 8.** *For all $m$ and $p, q$,*

$$\ell_1(Bin(m, p), Bin(m, q)) \geq \frac{1}{350} \min\left(m|p - q|, \frac{\sqrt{m}|p - q|}{\sqrt{p(1 - p)}}, 1\right).$$

*Proof.* For $m \leq 700$,

$$\ell_1(\mathrm{Bin}(m,p),\mathrm{Bin}(m,q)) \geq \ell_1(\mathrm{Ber}(p),\mathrm{Ber}(q)) = 2|p-q| \geq \frac{1}{350}\min\left(m|p-q|,\frac{\sqrt{m}|p-q|}{\sqrt{p(1-p)}},1\right),$$

Hence, in the rest of the proof, we focus on $m \geq 700$. Furthermore, since
$$\ell_1(\mathrm{Bin}(m,p),\mathrm{Bin}(m,q)) = \ell_1(\mathrm{Bin}(m,1-p),\mathrm{Bin}(m,1-q)).$$
and the bound $\frac{1}{350}\min\left(m|p-q|,\frac{\sqrt{m}|p-q|}{\sqrt{p(1-p)}},1\right)$ is symmetric in $p$ and $1-p$, it suffices to prove the result for $p \leq 1/2$.

Let $c = 2$. The proof for $mp \geq c$ is a direct consequence of Lemma 7. The proof for $c \leq 2$ follows from Lemma 6. $\square$

# D  Analysis of the algorithms

## D.1  Proof of Theorem 6

We first state the following guarantee on private hypothesis selection from Bun et al. [2019].

**Lemma 9** (Bun et al. [2019]). *Given $d$ distributions $p_1,p_2,\ldots,p_d$ and $n$ independent samples from an unknown distribution $p$, such that $\min_i \ell_1(p_i,p) \leq \alpha$, Algorithm 1 returns a distribution $p_i$ such that $\mathbb{E}[\ell_1(p_i,p)] \leq 4\alpha$, with probability $\geq 1-\beta$, if the number of samples satisfies,*

$$n \geq \frac{8\log(4m/\beta)}{\alpha^2} + \frac{8\log(4m/\beta)}{\alpha\varepsilon}.$$

*Furthermore, Algorithm 1 is $(\varepsilon,0)$-differentially private.*

*Proof.* The privacy guarantee follows by [Bun et al., 2019, Lemma 3.2]. The utility guarantee is obtained by applying the high probability utility bounds from [Bun et al., 2019, Lemma 3.3] and setting $\zeta = 1$. $\square$

Let $c$ be the constant in the lower bound of Theorem 5. Let $\mathcal{P} = \{0, \frac{c\alpha}{20m}, \frac{2c\alpha}{20m}, \ldots, 1]\}$ be a cover of $[0,1]$ Note that such that for every $p$, there exists a $p' \in \mathcal{P}$ such that

$$\min\left(m|p-p'|,\frac{\sqrt{m}|p-p'|}{\sqrt{p(1-p)}},1\right) \leq \frac{c\alpha}{10}.$$

Let $\mathcal{Q} = \{\mathrm{Bin}(m,p) : p \in \mathcal{P}\}$. Then by Theorem 5, for every $\mathrm{Bin}(m,p)$ there exists a $\mathrm{Bin}(m,p')$ in $\mathcal{Q}$ such that

$$\ell_1(\mathrm{Bin}(m,p),\mathrm{Bin}(m,p')) \leq \frac{c\alpha}{5}.$$

Hence, by Lemma 9, if

$$s = \Omega\left(\frac{8\log(20m/\alpha\beta)}{\alpha^2} + \frac{8\log(20m/\alpha\beta)}{\alpha\varepsilon}\right)$$

there is an algorithm that returns a distribution $\mathrm{Bin}(m,\hat{p}) \in \mathcal{Q}$ such that

$$\ell_1(\mathrm{Bin}(m,p),\mathrm{Bin}(m,\hat{p})) \leq \frac{4c\alpha}{5},$$

with probability $\geq 1-\beta$. Therefore, by the lower bound in Theorem 5, the resulting $\hat{p}$ satisfies

$$\min\left(m|p-\hat{p}|,\frac{\sqrt{m}|p-\hat{p}|}{\sqrt{p(1-p)}},1\right) \leq \frac{4\alpha}{5},$$

with probability $\geq 1-\beta$. Since $\frac{4\alpha}{5} \leq 1$, this implies that with probability $\geq 1-\beta$,

$$|p-\hat{p}| \leq \frac{4\alpha}{5}\max\left(\frac{1}{m},\frac{\sqrt{p(1-p)}}{\sqrt{m}}\right).$$

The expectation bound follows by setting $\beta = \alpha/5m$:

$$\mathbb{E}[|p-\hat{p}|] \leq \frac{4\alpha}{5}\max\left(\frac{1}{m},\frac{\sqrt{p(1-p)}}{\sqrt{m}}\right) + \frac{\alpha}{5m} \leq \alpha\max\left(\frac{1}{m},\frac{\sqrt{p(1-p)}}{\sqrt{m}}\right).$$

## D.2 Proof of Theorem 7

Let $\varepsilon' = \frac{\varepsilon}{4\sqrt{k \log \frac{1}{\delta}}}$ and $\alpha' = \min\left(\frac{\sqrt{m}\alpha}{2\sqrt{k}}, 1\right)$ We apply Theorem 6 for each symbol $k$ with $\varepsilon = \varepsilon'$ and $\alpha = \alpha'$ Then, we have an estimate $\hat{p}_1, \hat{p}_2, \ldots, \hat{p}_k$ such that

$$\mathbb{E}[\ell_1(p, \hat{p})] = \sum_i \mathbb{E}[|p_i - \hat{p}_i|]$$

$$\leq \alpha' \sum_i \max\left(\frac{1}{m}, \frac{\sqrt{p_i(1 - p_i)}}{\sqrt{m}}\right)$$

$$\leq \alpha' \sum_i \frac{1}{m} + \frac{\sqrt{p_i}}{\sqrt{m}}$$

$$\leq \frac{\alpha' k}{m} + \frac{\alpha' \sqrt{k}}{\sqrt{m}}$$

$$\leq 2\frac{\alpha' \sqrt{k}}{\sqrt{m}}$$

$$\leq \alpha,$$

where the penultimate follows from Jensen's inequality. The differential privacy bound follows from strong composition theorem (see [Kairouz et al., 2017, Theorem 3.4]) and using the fact that $e^{\varepsilon'} \leq 2\varepsilon'$.

## D.3 Proof of Lemma 2

Let $\hat{p}$ be such that

$$(1 - \hat{p})^m = \max\left(\min\left(\frac{1}{s}\sum_u 1_{N(u)=0} + \frac{Z}{s}, 1\right), 0\right), \tag{15}$$

Where $Z$ is a Laplace noise with parameter $1/\varepsilon$. Hence the algorithm is $(\varepsilon, 0)$-DP. Hence,

$$|(1 - \hat{p})^m - (1 - p)^m| \leq \left|\frac{1}{s}\sum_u 1_{N(u)=0} + \frac{Z}{s} - (1 - p)^m\right|.$$

Hence, by the tail bounds of the Laplace distribution, with probability $\geq 1 - 2\beta$,

$$|(1 - \hat{p})^m - (1 - p)^m| \leq \frac{\log \frac{1}{\beta}}{s\varepsilon} + \left|\frac{1}{s}\sum_u 1_{N(u)=0} - (1 - p)^m\right|.$$

Furthermore, by Bernstein's inequality with probability $\geq 1 - 2\beta$,

$$\left|\frac{1}{s}\sum_u 1_{N(u)=0} - (1 - p)^m\right| \leq 4\frac{\log \frac{1}{\beta}}{s} + 4\sqrt{\frac{\log \frac{1}{\beta}}{s} \cdot (1 - p)^m(1 - (1 - p)^m)}.$$

Since $1 - (1 - p)^m \leq mp$, we have with probability $\geq 1 - 4\beta$,

$$|(1 - \hat{p})^m - (1 - p)^m| \leq 4\sqrt{\frac{mp \log \frac{1}{\beta}}{s}} + 4\frac{\log \frac{1}{\beta}}{s} + \frac{\log \frac{1}{\beta}}{s\varepsilon}.$$

Combining with (14), with probability $\geq 1 - 4\beta$,

$$e^{-1.5c}\min(m|\hat{p} - p|, 0.5) \leq 4\sqrt{\frac{mp \log \frac{1}{\beta}}{s}} + 4\frac{\log \frac{1}{\beta}}{s} + \frac{\log \frac{1}{\beta}}{s\varepsilon}.$$

If $s \geq 64e^{3c}m \log \frac{3}{\beta}$, then the RHS is at most $e^{-1.5c}/2$. hence,

$$e^{-1.5c}m|\hat{p} - p| \leq 4\sqrt{\frac{mp \log \frac{1}{\beta}}{s}} + 4\frac{\log \frac{1}{\beta}}{s} + \frac{\log \frac{1}{\beta}}{s\varepsilon}.$$

If $s \geq \frac{16e^{3c}}{\alpha^2} \log \frac{3}{\beta} + \frac{16e^{3c}}{\gamma\varepsilon} \log \frac{3}{\beta}$

$$|p - \hat{p}| \leq \sqrt{\frac{p\alpha^2}{m}} + \frac{\alpha^2}{m} + \frac{\gamma}{m}.$$

## D.4 Proof of Theorem 8

**Parameters**: We first define few parameters. Let $\varepsilon' = \frac{\varepsilon}{8\sqrt{\min(k,m)\log\frac{1}{\delta}}}$, $\beta = \frac{\alpha}{40k}$, $\alpha' = \min\left(\frac{\sqrt{m}\alpha}{8\sqrt{k}}, 1\right)$, $\alpha'' = \frac{\alpha}{240}$, and $\gamma = \frac{m\alpha}{8k}$. Let $c = 4/m$.

**Algorithm**: For every symbol we first calculate the probability using the algorithm in Theorem 6 with $\varepsilon = \varepsilon'$, $\alpha = \alpha''$ and error probability $\beta$. If the estimated probability is less than $2/m$, we use the algorithm from Lemma 2 with $\varepsilon = \varepsilon'$, $\alpha = \alpha'$, $\gamma = \gamma$, and error probability $\beta$. Let $p'$ be the output of the first step and the $p''$ be the output of Lemma 2. The error of the algorithm is

$$|p - \hat{p}| = |p - p'|1_{p'>2/m} + |p - p''|1_{p'\leq 2/m}.$$

**Sample complexity**: The sample complexity would be the sum of sample complexities of Theorem 6 and Lemma 2 with appropriate parameters. Hence,

$$s \geq \frac{16\log(20m/\alpha''\beta)}{\alpha''^2} + \frac{16\log(20m/\alpha''\beta)}{\alpha''\varepsilon'} + \frac{16e^{3c}}{\alpha'^2}\log\frac{3}{\beta} + \frac{16e^{3c}}{\gamma\varepsilon'}\log\frac{3}{\beta}.$$

Hence, for a sufficient large constant $b$, if

$$s \geq b\log\frac{km}{\alpha} \cdot \left(\frac{k}{m\alpha^2} + \frac{k}{\sqrt{m}\varepsilon\alpha}\sqrt{\log\frac{1}{\delta}}\right)$$

Note that since $k \geq m$, the above bound implies that $s \geq b\sqrt{m}$, hence the bound also satisfies conditions in Lemma 2.

**Differential privacy:** We first provide the privacy guarantee for this algorithm. First observe that since $p', p'' \to \hat{p}$ is a Markov chain, by the postprocessing theorem it suffices to provide privacy guarantee for releasing $p', p''$. Consider releasing one of them, say $p'$. For any two neighboring datasets differ in at most $\min(m, k)$ symbols. Let these datasets be $D$ and $D'$ and $S(D, D')$ be the set of symbols where they differ. For these datasets,

$$\frac{\Pr(p'|D)}{\Pr(p'|D')} = \prod_{i\in S(D,D')}\frac{\Pr(p'_i|D)}{\Pr(p'_i|D')}.$$

Hence it suffices to apply strong composition theorem for this subset of size $\min(m, k)$ and the rest of the proof is similar to that of [Kairouz et al., 2017, Theorem 3.4]). The proof is similar for $p''$ and hence the result.

**Utility:** To analyze the utility, we divide the symbols into three sets $A_1 = \{i : p_i \geq \frac{4}{m}\}$, $A_2 = \{i : \frac{4}{m} \geq p_i \geq \frac{1}{4m}\}$, and $A_3 = \{i : p_i \leq \frac{1}{4m}\}$.

**Utility-large:** Consider the set $A_1$ with symbols whose probability is greater than $4/m$, for such a symbol, by Theorem 6, with probability $\geq 1 - \beta$,

$$|p - p'| \leq \alpha''\sqrt{\frac{p}{m}}.$$

Hence $p' \geq p - \alpha''\sqrt{\frac{p}{m}} > \frac{2}{m}$. Hence, for such a symbol with probability $\geq 1 - \beta$,

$$|p - \hat{p}| = |p - p'| \leq \alpha''\sqrt{\frac{p}{m}}.$$

**Utility-medium:** Consider the set $A_2$ with symbols whose probability in $[1/4m, 4/m]$. For such a symbol, then with probability $\geq 1 - 2\beta$,

$$|p - \hat{p}| \leq \max(|p - p'|, |p - p''|)$$
$$\leq \frac{2\alpha''}{m} + \alpha''\sqrt{\frac{p}{m}} + \alpha'\sqrt{\frac{p}{m}} + \frac{\alpha'^2}{m} + \frac{\gamma}{m}$$
$$\leq \frac{5\alpha''}{m} + \frac{\alpha'}{m} + \frac{\gamma}{m}.$$

**Utility-small:** Finally consider symbols whose probabilities are smaller than $1/4m$, for these symbols, with probability $\geq 1 - \beta$,

$$|p - p'| \leq \frac{\alpha''}{2m}.$$

and hence $p' \leq p + \frac{\alpha''}{m} \leq 3/2m \leq 2/m$. Hence only the second algorithm is used. Hence with probability $\geq 1 - 2\beta$, the error is at most,

$$|p - \hat{p}| = |p - p''| \leq \alpha'\sqrt{\frac{p}{m}} + \frac{\alpha'^2}{m} + \frac{\gamma}{m}.$$

Summing over all symbols yield,

$$
\begin{aligned}
\ell_1(p, \hat{p}) &\leq \sum_i |p_i - \hat{p}_i| \\
&\leq \sum_{i \in A_1} |p_i - \hat{p}_i| + \sum_{i \in A_2} |p_i - \hat{p}_i| + \sum_{i \in A_3} |p_i - \hat{p}_i| \\
&\leq \sum_{i \in A_1} \alpha''\sqrt{\frac{p_i}{m}} + \sum_{i \in A_2} \frac{5\alpha''}{m} + \frac{\alpha'}{m} + \frac{\gamma}{m} + \sum_{i \in A_3} \alpha'\sqrt{\frac{p}{m}} + \frac{\alpha'^2}{m} + \frac{\gamma}{m} \\
&\leq 28\alpha'' + \alpha'\left(\sqrt{\frac{k}{m}} + 1\right)\frac{k\alpha'^2}{m} + \frac{k\gamma}{m} + \\
&\leq \frac{\alpha}{8} + \frac{\alpha}{8} + \frac{\alpha}{8} + \frac{\alpha}{8} \\
&\leq \frac{\alpha}{2}.
\end{aligned}
$$

Hence, by the union bound, with probability with $1 - 20k\beta$,

$$\ell_1(p, \hat{p}) \leq \frac{\alpha}{2}.$$

Therefore in expectation,

$$\mathbb{E}[\ell_1(p, \hat{p})] \leq \frac{\alpha}{2} + 20k\beta \leq \alpha.$$

## E  Extensions

In this section, we modify our algorithms for the scenario when users have different number of samples. Let $m_{\max}$ be a known upper bound on the number of samples a user has. For a value $m$, let $s_m$ be the number of users such that $m_u \geq m$. Let $\bar{m}$ be the median values of $m_u$. We first state the main result, an analog of Theorem 2.

**Theorem 12.** *Let $\varepsilon \leq 1$. There exists a polynomial time algorithm $(\varepsilon, \delta)$-differentially private algorithm A such that*

$$S^A_{m,\alpha,\varepsilon,\delta} = \mathcal{O}\left(\log^2 \frac{km_{\max}}{\alpha} \cdot \max\left(\frac{k}{\bar{m}\alpha^2} + \frac{k}{\sqrt{\bar{m}}\alpha\varepsilon}\sqrt{\log \frac{1}{\delta}}, \frac{\sqrt{k}}{\varepsilon}\sqrt{\log \frac{1}{\delta}}\right)\right). \tag{16}$$

First we use $\varepsilon/2$ privacy budget find $\hat{m}$, a private estimate of $\bar{m}$, and $\hat{s}$, an estimate of $s_{\hat{m}}$ (the quantile of $\hat{m}$). We only keep the users with at least $\hat{m}$ samples, and select $\hat{m}$ samples from each of them. Hence we reduce the problem to the case when users have the same number of samples. Then we modify the algorithms for both the dense and sparse regimes so that they are differentially private even if the number of samples of a particular user changes. We use the remaining privacy budget for the modified algorithms. The privacy guarantee follows by the composition theorem.

We first provide the algorithm for privately estimating $\bar{m}$ and the quantile of estimated $\bar{m}$, which serves as a stepping stone for extending our algorithms to variable number of samples per user.

**Lemma 10.** *Let $s \geq \frac{16\log^2 m_{\max}/\beta}{\varepsilon}$. There exists a polynomial time $(\varepsilon, 0)$-algorithm that returns $\hat{m}$ and $\hat{s}$ such that with probability $\geq 1 - \beta$, the following holds,*

$$|\hat{s} - s_{\hat{m}}| \leq \frac{2\log^2 m_{\max}/\beta}{\varepsilon}, \quad \hat{m} \geq \frac{\bar{m}}{2}, \quad s_{\hat{m}} \geq \frac{s}{4}, \quad \hat{s} \geq \frac{3s}{8}. \tag{17}$$

*Proof.* Divide $\{0, 1, 2, \ldots, m_{\max}\}$ to bins $b_i$ such that $b_0 = 0$, $b_1 = 1$ and $b_i = 2 * b_{i-1}$ for $i \geq 1$. There are $v = \log m_{\max}$ buckets.

For any two adjacent datasets, $[t_0, t_1, t_2, \ldots, t_v]$ differ by two. Hence, we can add Laplace noise with parameter $\eta = 2/\varepsilon$ to each of them to obtain DP estimates. Let this be $[t'_0, t'_1, \ldots, t'_v]$.

By the tail bounds of Laplace distribution and the union bound, for each $i$ with probability $1 - \beta$,

$$|t_i - t'_i| \leq \eta \log \frac{v}{\beta}.$$

Furthermore, for any cumulative sets,

$$\left| \sum_{i \geq j} t_i - \sum_{i \geq j} t'_i \right| \leq \sum_{i \geq j} |t_i - t'_i| \leq \eta v \log \frac{v}{\beta}.$$

Let $j^*$ be the largest $j$ such that

$$\sum_{i \geq j} t'_i \geq \frac{s}{2} - \eta v \log \frac{v}{\beta}.$$

The algorithms return $\hat{s} = \sum_{i \geq j^*} t'_i$ and $\hat{m} = b_{j^*}$. Then by the assumption on $s$:

$$\hat{s} \geq \frac{s}{2} - \frac{2}{\varepsilon} \log m_{\max} \log \frac{\log m_{\max}}{\beta} \geq \frac{s}{2} - \frac{s}{8} = \frac{3s}{8}.$$

By the above cumulative equation sum,

$$|\hat{s} - s_{\hat{m}}| = \left| \sum_{i \geq j^*} t_i - \sum_{i \geq j^*} t'_i \right| \leq \eta v \log \frac{v}{\beta}.$$

$$s_{\hat{m}} = \sum_{i \geq j^*} t_i = \sum_{i \geq j^*} t'_i - \sum_{i \geq j^*} (t'_i - t_i) \geq \frac{s}{2} - \eta v \log \frac{v}{\beta} - \eta v \log \frac{v}{\beta} \geq \frac{s}{4}.$$

Note that by definition of $j^*$, $\sum_{i \geq j^*+1} t'_i < s/2 - v \log(v/\beta)$, and that $b_{j^*+1} = 2b_{j^*} = 2\hat{m}$, thus:

$$s_{2\hat{m}} = \sum_{i \geq j^*+1} t_i = \sum_{i \geq j^*+1} t'_i - \sum_{i \geq j^*+1} (t'_i - t_i) \leq \frac{s}{2} - \eta v \log \frac{v}{\beta} + \eta v \log \frac{v}{\beta} = \frac{s}{2}.$$

Hence $2\hat{m} \geq \bar{m}$. This completes the proof. $\qquad\square$

We proceed to discuss the algorithms for dense and sparse regimes. After we obtain $\hat{s}, \hat{m}$ from Lemma 10, we choose the algorithm depending on the relation between $k$ and $\hat{m}$: if $k \leq \hat{m}$, we use the algorithm for the dense regime; otherwise we use the one for the sparse regime.

### E.1 Dense regime

We first modify the hypothesis selection algorithm in Bun et al. [2019]. We cannot apply it directly because to ensure privacy, we cannot use the true number of users $s_{\hat{m}}$ and need to replace it with its private estimate $\hat{s}$. Hence we prove the following lemma to cope with this situation.

**Lemma 11.** *Let $\hat{s}, \hat{m}$ satisfy (17) with $\varepsilon = \varepsilon'$. Given $d$ distributions $p_1, p_2, \ldots, p_d$ and $s$ independent samples from an unknown distribution $p$, such that $\min_i \ell_1(p_i, p) \leq \alpha$, there exists an $(\varepsilon, 0)$-DP polynomial time algorithm that returns a distribution $p_i$ such that $\ell_1(p_i, p) \leq 4\alpha$, with probability $\geq 1 - \beta$, if the number of samples satisfies,*

$$s \geq \frac{128 \log^2(m_{\max}/\beta)}{3\alpha\varepsilon'} + \frac{32 \log(4d/\beta)}{\alpha^2} + \frac{64 \log(4d/\beta)}{3\alpha\varepsilon}.$$

*Proof.* Let $H$ and $H'$ be two distributions over the domain $\mathcal{X}$ and define the Scheffe set

$$\mathcal{W}_1 = \{x \in \mathcal{X} : H(x) > H'(x)\}.$$

Define $p_1 = H(\mathcal{W}_1), p_2 = H'(\mathcal{W}_1)$, for some distribution $P$ define $\tau = P(\mathcal{W}_1)$. Note that $p_1 > p_2$ and $p_1 - p_2 = d_{TV}(H, H')$.

Let $D$ be a dataset of size $s_{\hat{m}}$ drawn i.i.d. from $P$. Define the following quantities which serve as empirical estimates of $P(\mathcal{W}_1)$,

$$\hat{P}(\mathcal{W}_1) := \hat{\tau} := \frac{1}{\hat{s}}|\{x \in D : x \in \mathcal{W}_1\}|, \quad P_{\hat{m}}(\mathcal{W}_1) := \tau_{\hat{m}} := \frac{1}{s_{\hat{m}}}|\{x \in D : x \in \mathcal{W}_1\}|.$$

Let $\zeta > 0$ be the approximation parameter. Consider the function

$$\hat{\Gamma}_\zeta(H, H', D) = \begin{cases} \hat{s} & p_1 - p_2 \leq (2+\zeta)\alpha; \\ \hat{s} \cdot \max\{0, \hat{\tau} - (p_2 + (1+\zeta/2)\alpha)\} & \text{otherwise.} \end{cases}$$

According to [Bun et al., 2019, Lemma 3.1, Lemma 3.3], $\hat{\Gamma}_\zeta$ has the following properties,

**Lemma 12** (Bun et al. [2019], Lemma 3.1)**.** *If $d_{TV}(P, H) \leq \alpha$ and $|\hat{\tau} - \tau| < \zeta\alpha/4$, then $\hat{\Gamma}_\zeta(H, H', D) > \zeta\alpha\hat{s}/4$.*

**Lemma 13** (Bun et al. [2019], Lemma 3.3)**.** *If $d_{TV}(P, H') \leq \alpha$ , $|\hat{\tau} - \tau| < \zeta\alpha/4$, and $\hat{\Gamma}_\zeta(H, H', D) > 0$, then $d_{TV}(H, H') \leq (2+\zeta)\alpha$.*

Define the score functions for each $H_j \in \mathcal{H}$

$$\hat{S}(H_j, D) = \min_{H_k \in \mathcal{H}} \hat{\Gamma}_\zeta(H_j, H_k, D).$$

Output a random hypothesis $\hat{H}$ according to the distribution

$$\Pr[\hat{H} = H_j] \propto \exp\left(\frac{\hat{S}(H_j, D)}{2\varepsilon}\right).$$

First note that if $d_{TV}(P, H) < \alpha$, then using Hoeffding's inequality, we have with probability at least $1 - 2\exp(-s_{\hat{m}}\zeta^2\alpha^2/32)$,

$$|\tau_{\hat{m}} - \tau| < \zeta\alpha/8.$$

Assume that there exists $H^* \in \mathcal{H}$ such that $d_{TV}(P, H^*) \leq \alpha$. Define $\mathcal{W}_j = \{x \in \mathcal{X} : H^*(x) > H_j(x)\}$. Conditioned on that the inequalities in Lemma 10 hold, by the union bound, with probability at least $1 - 2d\exp(-s_{\hat{m}}\zeta^2\alpha^2/8) \geq 1 - 2d\exp(-s\zeta^2\alpha^2/32)$ over the draws of $D$, for all $j$ we have

$$|P(\mathcal{W}_j) - P_{\hat{m}}(\mathcal{W}_j)| \leq \zeta\alpha/8.$$

Due to the inequalities in Lemma 10, the following holds uniformly for all $j$,

$$|\hat{P}(\mathcal{W}_j) - P_{\hat{m}}(\mathcal{W}_j)| \leq \left|\frac{1}{\hat{s}} - \frac{1}{s_{\hat{m}}}\right| s_{\hat{m}} = \frac{|\hat{s} - s_{\hat{m}}|}{\hat{s}} \leq \frac{16\log^2(m_{\max}/\beta)}{3s\varepsilon'}$$

Hence as long as $s > \frac{128\log^2(m_{\max}/\beta)}{3\zeta\alpha\varepsilon'}$, the above quantity is bounded by $\zeta\alpha/8$. We have

$$|P(\mathcal{W}_j) - \hat{P}(\mathcal{W}_j)| \leq |P(\mathcal{W}_j) - P_{\hat{m}}(\mathcal{W}_j)| + |\hat{P}(\mathcal{W}_j) - P_{\hat{m}}(\mathcal{W}_j)| \leq \frac{\zeta\alpha}{4}.$$

By Lemma 12 we have $\hat{\Gamma}_\zeta(H^*, H_j, D) > \zeta\alpha\hat{s}/4 \geq 3\zeta\alpha s/32$. This implies $\hat{S}(H^*, D) > 3\zeta\alpha s/32$.

By the utility of the exponential mechanism, with probability at least $1 - \beta/2$, the output hypothesis $\hat{H}$ satisfies

$$\hat{S}(\hat{H}, D) \geq \hat{S}(H^*, D) - \frac{2\log(2d/\beta)}{\varepsilon}$$
$$\geq \frac{3\zeta\alpha s}{32} - \frac{2\log(2d/\beta)}{\varepsilon}.$$

As long as $s \geq \frac{32\log(4d/\beta)}{\zeta^2\alpha^2} + \frac{64\log(2d/\beta)}{3\zeta\alpha\varepsilon}$, together with probability at least $1 - \beta$ , $\hat{S}(\hat{H}, D) > 0$, which implies that $\hat{\Gamma}_\zeta(\hat{H}, H^*, D) > 0$. Since in addition $d_{TV}(P, H^*) \leq \alpha$, we have $d_{TV}(\hat{H}, H^*) \leq (2+\zeta)\alpha$ by Lemma 13 and hence $d_{TV}(\hat{H}, P) \leq (3+\zeta)\alpha$. Setting $\zeta = 1$ gives the desired result. $\quad\square$

**Theorem 13.** *Suppose there are $s$ users such that user $u$ has $m_u$ i.i.d. samples from $\mathrm{Ber}(p)$. Let $\hat{s}, \hat{m}$ satisfy (17) with $\varepsilon = \varepsilon'$. Let $s \geq \frac{128 \log^2(m_{\max}/\beta)}{3\alpha\varepsilon'} + \frac{32 \log(80m_{\max}/\alpha\beta)}{\alpha^2} + \frac{64 \log(80m_{\max}/\alpha\beta)}{3\alpha\varepsilon}$. There exists a polynomial time $(\varepsilon, 0)$ differentially private algorithm that returns $\hat{p}$ such that with probability at least $1 - \beta$,*

$$|p - \hat{p}| \leq \frac{4}{5}\alpha \max\left(\frac{1}{\hat{m}}, \frac{\sqrt{p(1-p)}}{\sqrt{\hat{m}}}\right).$$

*Proof.* We sample $\hat{m}$ samples from all users that have least $\hat{m}$ samples. Hence we obtain $s_{\hat{m}}$ i.i.d samples from $\mathrm{Bin}(\hat{m}, p)$. Let $c$ be the constant in Theorem 5. We then apply the modified hypothesis selection algorithm in Lemma 11 with the hypothesis class $\mathcal{Q} = \{\mathrm{Bin}(\hat{m}, p), p \in \mathcal{P}\}$ where $\mathcal{P} = \{0, \frac{c\alpha}{20\hat{m}}, \frac{2c\alpha}{20\hat{m}}..., 1\}$. The total number of hypotheses is $d = \frac{20\hat{m}}{c\alpha}$. The sample complexity comes from Lemma 11 and utility follows by the argument in Theorem 6 with $m$ replaced by $\hat{m}$.

By Theorem 5, for every $\mathrm{Bin}(\hat{m}, p)$ there exists a $\mathrm{Bin}(\hat{m}, p')$ in $\mathcal{Q}$ such that

$$\ell_1(\mathrm{Bin}(\hat{m}, p), \mathrm{Bin}(\hat{m}, p')) \leq \frac{c\alpha}{5}.$$

Hence, by Lemma 11, if

$$s = \Omega\left(\frac{128\log^2(m_{\max}/\beta)}{3\alpha\varepsilon'} + \frac{32\log(80m_{\max}/\alpha\beta)}{\alpha^2} + \frac{64\log(80m_{\max}/\alpha\beta)}{3\alpha\varepsilon}\right),$$

there is an algorithm that returns a distribution $\mathrm{Bin}(m, \hat{p}) \in \mathcal{Q}$ such that

$$\ell_1(\mathrm{Bin}(\hat{m}, p), \mathrm{Bin}(\hat{m}, \hat{p})) \leq \frac{4c\alpha}{5},$$

with probability $\geq 1 - \beta$. Therefore, by the lower bound in Theorem 5, the resulting $\hat{p}$ satisfies

$$\min\left(\hat{m}|p - \hat{p}|, \frac{\sqrt{\hat{m}}|p - \hat{p}|}{\sqrt{p(1-p)}}, 1\right) \leq \frac{4\alpha}{5},$$

with probability $\geq 1 - \beta$. Since $\frac{4\alpha}{5} \leq 1$ and $\hat{m} \geq \bar{m}/2$, this implies that with probability $\geq 1 - \beta$,

$$|p - \hat{p}| \leq \frac{4\alpha}{5}\max\left(\frac{1}{\hat{m}}, \frac{\sqrt{p(1-p)}}{\sqrt{\hat{m}}}\right) \leq \frac{4\alpha}{5}\max\left(\frac{2}{\bar{m}}, \frac{\sqrt{2p(1-p)}}{\sqrt{\bar{m}}}\right).$$

The expectation bound follows by setting $\beta = \alpha/5m_{\max}$,

$$\mathbb{E}[|p - \hat{p}|] \leq \frac{4\alpha}{5}\max\left(\frac{2}{\bar{m}}, \frac{\sqrt{2p(1-p)}}{\sqrt{\bar{m}}}\right) + \frac{\alpha}{5m_{\max}} \leq \alpha\max\left(\frac{2}{\bar{m}}, \frac{\sqrt{2p(1-p)}}{\sqrt{\bar{m}}}\right).$$

$\square$

**Theorem 14** (Dense regime). *Let $k \leq \hat{m}$ and $\varepsilon \leq 1$. There exists a polynomial time $(\varepsilon, \delta)$-differentially private algorithm $A$ such that*

$$S^A_{m,\alpha,\varepsilon,\delta} = \mathcal{O}\left(\log^2\frac{km_{\max}}{\alpha} \cdot \max\left(\frac{k}{\bar{m}\alpha^2} + \frac{k}{\sqrt{\bar{m}}\alpha\varepsilon}\sqrt{\log\frac{1}{\delta}}, \frac{\sqrt{k}}{\varepsilon}\sqrt{\log\frac{1}{\delta}}\right)\right).$$

*Proof.* Let $\beta > 0$ be the probability guarantee to be chosen later. Use $\varepsilon_1 = \varepsilon/2$ budget to obtain $\hat{s}, \hat{m}$ using Lemma 10, which satisfy (17) with probability at least $1 - \beta$ as long as $s \geq \frac{16\log^2 m_{\max}/\beta}{\varepsilon/2}$.

Define $\varepsilon_2 = \frac{\varepsilon}{8\sqrt{(k+1)\log(2/\delta)}}, \alpha' = \min\left(\frac{\sqrt{\bar{m}}\alpha}{2\sqrt{k}}, 1\right)$. Under the condition above, by union bound and applying Theorem 13 with $\varepsilon' = \varepsilon_1, \varepsilon = \varepsilon_2, \alpha = \alpha'$, with probability at least $1 - k\beta$, for all $\hat{p}_i$ we have

$$|p_i - \hat{p}_i| \leq \frac{4}{5}\alpha' \max\left(\frac{1}{\hat{m}}, \frac{\sqrt{p(1-p)}}{\sqrt{\hat{m}}}\right),$$

as long as

$$s = \Omega\left(\log^2 \frac{m_{\max}}{\alpha\beta} \max\left(\frac{k}{\bar{m}\alpha^2} + \frac{\sqrt{k}}{\alpha\varepsilon_2\sqrt{\bar{m}}}, \frac{1}{\varepsilon_2}\right)\right) \tag{18}$$

$$\geq \frac{128\log^2(m_{\max}/\beta)}{3\alpha'\varepsilon_1} + \frac{32\log(80m_{\max}/\alpha'\beta)}{(\alpha')^2} + \frac{64\log(80m_{\max}/\alpha'\beta)}{3\alpha'\varepsilon_2}.$$

Note that this satisfies the condition on $s$ in Lemma 10. Together with probability at least $1 - (k+1)\beta$:

$$\ell_1(p, \hat{p}) = \sum_i |p_i - \hat{p}_i|$$

$$\leq \frac{4}{5}\alpha' \sum_i \max\left(\frac{1}{\hat{m}}, \frac{\sqrt{p_i(1-p_i)}}{\sqrt{\hat{m}}}\right)$$

$$\leq \frac{4}{5}\alpha' \sum_i \frac{1}{\hat{m}} + \frac{\sqrt{p_i}}{\sqrt{\hat{m}}}$$

$$\leq \frac{4}{5}\left(\frac{\alpha' k}{\hat{m}} + \frac{\alpha'\sqrt{k}}{\sqrt{\hat{m}}}\right)$$

$$\leq 2\frac{4}{5}\frac{\alpha'\sqrt{k}}{\sqrt{\hat{m}}}$$

$$\leq \frac{4}{5}\alpha,$$

Choosing $\beta = \frac{\alpha}{40k}$,

$$\mathbb{E}[\ell_1(\hat{p}, p)] \leq \frac{4\alpha}{5} + 2(k+1)\beta = \alpha.$$

Plug in $\varepsilon_2$ and $\beta$ in (18) we obtain the desired user complexity. Privacy guarantee follows by the composition theorem. □

## E.2 Sparse regime

**Lemma 14.** *Let* $\hat{s}, \hat{m}$ *satisfy* (17) *with* $\varepsilon = \varepsilon'$. *Let* $p \leq \min(c/\hat{m}, 1/2)$. *Let* $s \geq 64e^{3c}\max(c,1)\log\frac{3}{\beta}$ *and* $s \geq \frac{128e^{3c}}{\alpha^2}\log\frac{3}{\beta} + \frac{32e^{3c}}{\gamma\varepsilon'}\log^2\frac{3m_{\max}}{\beta} + \frac{16e^{3c}}{\gamma\varepsilon}\log\frac{3}{\beta}$. *There exists a polynomial time* $(\varepsilon, \delta)$*-estimator* $\hat{p}$ *such that with probability at least* $1 - \beta$,

$$|p - \hat{p}| \leq \sqrt{\frac{p\alpha^2}{\hat{m}}} + \frac{\alpha^2}{\hat{m}} + \frac{\gamma}{\hat{m}\varepsilon}.$$

*Proof.* We modify the algorithm for the sparse regime as follows.

Let $\mathcal{U}_{\hat{m}}$ be the users who have at least $\hat{m}$ samples. Similar to (15), we find $\hat{p}$ such that,

$$(1 - \hat{p})^{\hat{m}} = \max\left(\min\left(\frac{1}{\hat{s}}\sum_{u\in\mathcal{U}_{\hat{m}}} 1_{N(u)=0} + \frac{Z}{\hat{s}}, 1\right), 0\right),$$

where $Z = Lap(1/\varepsilon)$. Therefore,

$$|(1-\hat{p})^{\hat{m}} - (1-p)^{\hat{m}}| \leq \left|\frac{1}{\hat{s}}\sum_{u\in\mathcal{U}_{\hat{m}}} 1_{N(u)=0} + \frac{Z}{\hat{s}} - (1-p)^{\hat{m}}\right|$$

$$\leq \left|\frac{1}{\hat{s}}\sum_{u\in\mathcal{U}_{\hat{m}}} 1_{N(u)=0} - \frac{1}{s_{\hat{m}}}\sum_{u\in\mathcal{U}_{\hat{m}}} 1_{N(u)=0}\right| + \frac{|Z|}{\hat{s}}$$

$$+ \left|\frac{1}{s_{\hat{m}}}\sum_{u\in\mathcal{U}_{\hat{m}}} 1_{N(u)=0} - (1-p)^{\hat{m}}\right|.$$

From Lemma 10, with probability at least $1 - \beta$, the first term is upper bounded by

$$\left| \frac{1}{\hat{s}} - \frac{1}{s_{\hat{m}}} \right| \left| \sum_{u \in \mathcal{U}_{\hat{m}}} 1_{N(u)=0} \right| \leq \left| \frac{\hat{s} - s_{\hat{m}}}{\hat{s}} \right| \leq \frac{16 \log^2(m_{\max}/\beta)}{3s\varepsilon'}.$$

The second and third term are bounded similar to Lemma 2 using Laplace tail bounds and Bernstein's inequality. With probability $1 - 4\beta$,

$$\frac{|Z|}{\hat{s}} + \left| \frac{1}{s_{\hat{m}}} \sum_{u \in \mathcal{U}_{\hat{m}}} 1_{N(u)=0} - (1-p)^{\hat{m}} \right| \leq \frac{\log(1/\beta)}{\hat{s}\varepsilon} + 4\sqrt{\frac{\hat{m}p \log \frac{1}{\beta}}{s_{\hat{m}}}} + 4\frac{\log \frac{1}{\beta}}{s_{\hat{m}}}.$$

Together with probability at least $1 - 5\beta$,

$$e^{-1.5c} \min\{\hat{m}|\hat{p} - p|, 0.5\} \leq |(1 - \hat{p})^{\hat{m}} - (1 - p)^{\hat{m}}|$$

$$\leq \frac{16 \log^2(m_{\max}/\beta)}{3s\varepsilon'} + \frac{\log(1/\beta)}{\hat{s}\varepsilon} + 4\sqrt{\frac{\hat{m}p \log \frac{1}{\beta}}{s_{\hat{m}}}} + 4\frac{\log \frac{1}{\beta}}{s_{\hat{m}}}$$

$$\leq \frac{16 \log^2(m_{\max}/\beta)}{3s\varepsilon'} + \frac{8 \log(1/\beta)}{3s\varepsilon} + 8\sqrt{\frac{\hat{m}p \log \frac{1}{\beta}}{s}} + \frac{16 \log \frac{1}{\beta}}{s}.$$

The last inequality is due to $\hat{s} \geq 3s/8$ and $s_{\hat{m}} \geq s/4$.

If $s \geq 256 e^{3c} p\hat{m} \log(3/\beta)$, then the right hand side is upper bounded by $e^{-1.5c}/2$. Thus,

$$e^{-1.5c}\hat{m}|\hat{p} - p| \leq \frac{16 \log^2(m_{\max}/\beta)}{3s\varepsilon'} + \frac{8 \log(1/\beta)}{3s\varepsilon} + 8\sqrt{\frac{\hat{m}p \log \frac{1}{\beta}}{s}} + \frac{16 \log \frac{1}{\beta}}{s}.$$

If $s \geq \frac{128 e^{3c}}{\alpha^2} \log \frac{3}{\beta} + \frac{32 e^{3c}}{\gamma \varepsilon'} \log^2 \frac{3m_{\max}}{\beta} + \frac{16 e^{3c}}{\gamma \varepsilon} \log \frac{3}{\beta}$,

$$|\hat{p} - p| \leq \sqrt{\frac{p\alpha^2}{\hat{m}}} + \frac{\alpha^2}{\hat{m}} + \frac{\gamma}{\hat{m}}.$$

In the end we get a result similar to Lemma 3. $\qquad\square$

**Theorem 15.** *Let $\varepsilon \leq 1$ and $k \geq \hat{m}$. There exists a polynomial time $(\varepsilon, \delta)$-differentially private algorithm A such that*

$$S^A_{m,\alpha,\varepsilon,\delta} = \mathcal{O}\left( \log^2 \frac{km_{\max}}{\alpha} \cdot \left( \frac{k}{\bar{m}\alpha^2} + \frac{k}{\sqrt{\bar{m}}\varepsilon\alpha} \sqrt{\log \frac{1}{\delta}} \right) \right).$$

*Proof.* Like the algorithm for the dense regime, we first use $\varepsilon_1 = \frac{\varepsilon}{2}$ budget to estimate $\hat{s}, \hat{m}$. Then we define the following parameters,

$$\varepsilon_2 = \frac{\varepsilon/2}{8\sqrt{\min(k,\hat{m}) \log \frac{1}{\delta/2}}}, \; \beta = \frac{\alpha}{40k}, \; \alpha' = \min\left( \frac{\sqrt{\hat{m}}\alpha}{8\sqrt{k}}, 1 \right), \; \alpha'' = \frac{\alpha}{240}, \; \gamma = \frac{\hat{m}\alpha}{8k}$$

The proof follows similarly as Theorem 8.

**Algorithm**: For every symbol we first calculate the probability using the algorithm in Theorem 13 with $\varepsilon' = \varepsilon_1, \varepsilon = \varepsilon_2, \alpha = \alpha''$ and error probability $\beta$. If the estimated probability is less than $2/\hat{m}$, we use the algorithm from Lemma 14 with $\varepsilon' = \varepsilon_1, \varepsilon = \varepsilon_2, \alpha = \alpha', \gamma = \gamma$, and error probability $\beta$. Let $p'$ be the output of the first step and the $p''$ be the output of Lemma 14. The error of the algorithm is

$$|p - \hat{p}| = |p - p'|1_{p' > 2/m} + |p - p''|1_{p' \leq 2/m}.$$

**Sample complexity**: The sample complexity would be the sum of sample complexities of Theorem 13 and Lemma 14 with appropriate parameters. Hence,

$$s \geq \frac{128 \log^2(m_{\max}/\beta)}{3\alpha''\varepsilon_1} + \frac{32 \log(80m_{\max}/c\alpha''\beta)}{\alpha''^2} + \frac{64 \log(80m_{\max}/c\alpha''\beta)}{3\alpha''\varepsilon_2}$$
$$+ \frac{128e^{3c}}{\alpha'^2} \log \frac{3}{\beta} + \frac{32e^{3c}}{\gamma\varepsilon_1} \log^2 \frac{3m_{\max}}{\beta} + \frac{16e^{3c}}{\gamma\varepsilon_2} \log \frac{3}{\beta}.$$

Hence, for a sufficient large constant $b$, if

$$s \geq b \log^2 \frac{km_{\max}}{\alpha} \cdot \left( \frac{k}{\bar{m}\alpha^2} + \frac{k}{\sqrt{\bar{m}}\varepsilon\alpha} \sqrt{\log \frac{1}{\delta}} \right).$$

Note that since $k \geq \hat{m}$, the above bound implies that $s \geq b\sqrt{\hat{m}}$, hence the bound also satisfies conditions in Lemma 14 and Lemma 10.

Following the same argument as Theorem 8, the algorithm after we obtain $\hat{s}, \hat{m}$ is $(\varepsilon/2, \delta/2)$ private. Using the naive composition theorem, the entire algorithm is $(\varepsilon, \delta)$ private.

Utility follows by the argument in Theorem 8 with $m$ replaced by $\hat{m}$. $\qquad\square$