[Reviews · NeurIPS 2020]

Review 1

Summary and Contributions: The authors consider the problem of learning discrete distribution concerning user-level differential privacy. The setting of the problem is as follows: Each user receives m iid samples from an unknown distribution p over [k], and the goal is to learn p in \ell_1 distance using as few users as possibles. The privacy guarantee is provided in a particular manner: In the standard differential privacy setting, the idea is to ensure that the algorithm behavior does not change drastically if we change one sample (item). However, for the user-level privacy, while each user may have several data points, we would like the algorithm's behavior remains almost the same if the data points of a user change. The authors provide almost matching upper and lower bounds for the problem. They have extended their result to the case where users receive different numbers of samples. They also discuss the limitation of the following naïve approach for solving the problem: One can take the empirical distribution from the aggregated data of all the users and use exponential or Gaussian mechanisms to privatize it. However, the authors proved that any algorithm which uses the aggregated data count would incur sub-optimal sample complexity. In the proof, they introduce the notion of restricted estimators and provide a coupling argument to establish this lower bound.

Strengths: The problem of learning discrete distribution is fundamental in statistics and theoretical computer science. Studying this problem from a novel perspective in differential privacy is an important task. The authors obtained a nearly optimal algorithm for this problem with compelling ideas. The arguments are explained clearly, and the paper is well-written.

Weaknesses: - In the paper, the goal is to bound the expected \ell_1 error as the loss function. In contrast, in the PAC learning model, we are looking for the worst-case \ell_1 error, which is guaranteed with some high probability 1 - \delta (known as the confidence parameter). It would be interesting to discuss the relationship between the two models. In particular, for a constant confidence parameter, the results of this submission can be easily extended to the PAC learning model. Obtaining an algorithm with the right dependency to \delta might be more challenging. What are the advantages of the expected \ell_1 error? Does it simplify the proof? - There is no experiment that echos the main result of the paper. An experiment is provided to validate an auxiliary theorem (Theorem 5), which is about learning multinomial distributions.

Correctness: Based on the main text, the theoretical results seem correct.

Clarity: The paper is very well-written.

Relation to Prior Work: Related work is properly discussed to the best of my knowledge. As it is mentioned in the paper, the coupling technique for the differentially private lower bounds has been used before [Acharya et al. 2018, and Acharya et al. 2020]. Theorem 4 extends those results to restricted estimators. However, it is unclear whether the technical ideas in the proof differentiates drastically from the previous work.

Reproducibility: Yes

Additional Feedback: Minor comments: Line 99 to Line 108: N is used in three different ways: N(u), N(u, D), and N(D). Line 225: The authors might want to consider changing \delta to something else. =============== Update ================= Thank you for your response. I hope your paper gets in.


Review 2

Summary and Contributions: This work considers the problem of learning discrete distributions form the point of view of user level privacy, a notion of privacy that, somewhat differently than differential privacy, requires the stronger property that one preserve privacy for all items of a single user. One assumes a bound m on the number of items (samples) given to each user. Upper bounds on the sample complexity are given for the Laplace and Gaussian mechanisms. Lower bounds on a wide class of estimators (which rely only on the final count) are given -- since this class include Laplace/Gaussian, new algorithmic ideas are necessary. In fact, they give such a new algorithm with improved sample complexity (beating their first lower bound). Their algorithm breaks down the problem into two cases, depending on the relationship between the number of samples per user m and the size of the domain k. Finally they give an information theoretic lower bound on differentially private algorithms, showing that their algorithm is close to optimal.

Strengths: Privacy is an important domain. The notion of privacy is interesting.

Weaknesses: Unclear whether there is novelty in the techniques, but that is ok.

Correctness: yes

Clarity: yes

Relation to Prior Work: yes

Reproducibility: Yes

Additional Feedback:


Review 3

Summary and Contributions: The paper considers the problem of privately estimating a discrete distribution of items (e.g., emoji frequency) when each individual can be contributing multiple items to the dataset. When each individual can produce m-samples at most, the paper establishes upper and lower bounds of the number of individuals needed to achieve \alpha-accurate estimation in terms of TV-distance. The author shows that one can beat the baseline approaches of Gaussian / Laplace mechanism in two specialized regimes. The improvement is in the term of the part of the sample (user) complexity due to privacy constraints by a factor of \sqrt{m} where m is the maximum number of samples per user.

Strengths: - The lower bounds constructions seem to involve some interesting new tools that can be used more generically. Although the main contributions there seem to have already been presented in the cited work by Acharaya et al. 2018; 2020. It will help us to evaluate the technical novelty if the authors comment on the challenges in applying those techniques in the context of constructing the presented lower bounds. - The algorithm for the upper bound involve new algorithmic insight.

Weaknesses: - The result for the proposed algorithm seems to require an additional assumption that each individual’s data is iid drawn from the same distribution. Otherwise I don’t see how that \sqrt{m} argument in the beginning of Section 5.1 works out and how Theorem 6 can be applied to prove Theorem 7. I find such assumption unjustifiable because in practice, each users’ preferred set of “emojis” are very different. - It is unreasonable to assume each user contributes m data points. Usually users are very heterogeneous in terms of the number of data points they produce. This affects the user-complexity calculations and makes the problem more interesting. In the worst case, m can be as large as n, still with appropriate truncation, one can obtain meaningful frequency estimates with data-dependent privacy-mechanisms and utility bounds. Unfortunately this is not considered in the paper. - Han et al that the authors cited for the lower bounds of non-private estimation of L1-distance establishes an adaptive (upper and lower) bound that depends on the entropy of the distribution. While this, in the worst case, is proportional to k, it is often much smaller. The results in this paper do not seem to concern dependence on k. Replacing k with the entropy in the term that is introduced by DP will make the result much more interesting. - The research in estimating discrete distributions have evolved quite a bit nowadays. Other loss functions are of interest too, e.g., KL-divergence, \chi-square distance. Those metrics sometimes allow for more interesting rates and more interesting worst-case dependence on k (often log k). There are competitive notions of optimality that I encourage the authors to look into, see, e.g.: Orlitsky, A., & Suresh, A. T. (2015). Competitive distribution estimation: Why is good-turing good. In Advances in Neural Information Processing Systems (pp. 2143-2151).

Correctness: The results seem correct to the best of my ability (and time constraint) to check --- **provided that** the additional assumption from my first comment in “Weaknesses” is made. Without that, I think it is not possible to apply Theorem 6, because you do not have multiple samples from the same unknown binomial distribution that are independent.

Clarity: The paper is generally easy to follow, but there are room to improve on readability and self-containedness. - The authors should consider adding a pseudo-code block. Saying only “By proposing a modified Scheffe estimator ” is not enough to inform readers on how exactly this estimator works and how it leverages the intuition that was presented at the beginning of Section 5.1. Also, a proof sketch of Theorem 6 and Theorem 7 is probably best presented in the main paper. - Theorem 6 seems to require the s samples to be independent? - s is not defined in Lemma 3. It is unclear what s is and how is it related to the \hat{p}. I managed to figure out the answers by reading the proof but it wasn’t clear from the statements. There are sufficient number of such issues throughout the paper that could make it a painful reading experience for people who have less expertise in either statistical estimation or in differential privacy. I strongly encourage the authors to give the manuscript an extra editing pass and potentially get some non-authors to read and provide feedback.

Relation to Prior Work: The discussion of related work is passable for a conference paper submission. There are obviously many problems that share related structures that have been studied extensively in DP literature, e.g., nodal DP for releasing graphs, e.g., post-processing approaches to get exponential improvements in histogram estimations in sparse regimes and so on. They are not directly subsuming the current work, so it is OK to not refer to them. I think there are interesting lessons that can be learned from there for the current problem.

Reproducibility: Yes

Additional Feedback: Additional feedback, places to improve 1. Addressing my first concern in the weaknesses will make the paper more applicable to the motivating applications. I would consider increasing my score if it is addressed satisfactorily in the rebuttal. 2. Can the authors discuss the exponential dependence on c in Lemma 3? 3. What happens to the many cases in between the dense and sparse regime? Do you get better bounds than the Laplace / Gaussian mechanism? 4. The setting that applies to arbitrary data sets when individual constributions are not constrained a priori is going to make the paper more interesting. This is related to the problem of nodal differential privacy in differentially private release of graphs. The authors should check out the Lipschitz extensions and it may help the authors for this purposes. See: Kasiviswanathan, S. P., Nissim, K., Raskhodnikova, S., & Smith, A. (2013, March). Analyzing graphs with node differential privacy. In Theory of Cryptography Conference (pp. 457-476). Springer, Berlin, Heidelberg. ------------------- updates after author feedback ------------ I appreciate the authors responding in details. Thanks for pointing out the results that depend on the median. Regarding the iid data assumption (within user and across user), the authors provided a number of references under this assumption that study this problem under item-level DP. I argue that on the item-level, this is a reasonable assumption because if you randomly shuffle the observations, then everything you see remains identical. If this assumption is not true, then it does not have a strong impact in the accuracy guarantee of the LDP releases. On the user-level, it becomes somewhat unreasonable, particularly because this is **the very information** that the algorithm proposed by the authors leverage upon. If this assumption fails --- which is always the case --- then the proposed algorithm may fail completely. Moreover, even as a purely theoretical study, the heterogeneity of the users is arguably the main challenge in the first place. While I agree that the paper is a solid first step towards addressing user-level privacy, the model assumption and discussions of its pros and cons should be made clear up front in the paper. Regarding the authors comment that the analysis would give insights to "design algorithms that might perform well in the non iid setting". The authors could try to either improve the upper bound or the lower bound (Theorem 3). my conjecture is that under the distinct distribution setting, it is the lower bound that could be improved and you need some parameter to discribe the extent to which the algorithmic insight from this paper can be applied. At the moment, the proposed algorithm may lead to worse error in the non-iid setting as far as I can tell, so designing algorithms that exploit such information adaptively, without relying strongly on the assumption would be the key. In the likely event that the paper is accepted, I encourage the authors to discuss these subtleties.


Review 4

Summary and Contributions: This paper combines two recent threads in the theory literature: user-level differential privacy and estimation of discrete distributions. As a quick summary, the sample complexity of discrete distribution estimation has been sharply characterized as well as the item-level differential privacy equivalent. Item-level privacy measures privacy at the level of individual elements of the database whereas user-level privacy assumes that each user contributes several items to the database, and privacy must be afforded to all of a user's items simultaneously. A naive distribution estimator must ignore all but one of the items from each user in order to maintain privacy. This paper proposes a novel estimator that can benefit from multiple items per user, thus reducing the privacy penalty substantially, and also proposes a matching information-theoretic lower bound. There is no empirical component.

Strengths: This is a solid theoretical paper that 1) proposes an interesting and novel generalization on a problem of recent interest, 2) demonstrates that existing approaches such as Gaussian or Laplacian noise mechanism (and any variants that rely only on count statistics) are not up to the task, 3) proposes a novel estimator that outperforms prior work, and 4) proves a matching lower bound. The analysis requires some interesting generalizations on Assouad's bound.

Weaknesses: Edit after author response: I think that the presentation could be slightly improved to both mention scenarios where the single distribution is relevant (e.g., word prediction as you suggested) and that distinct distributions is an interesting direction to generalize your bounds. That is, I would be careful not to "oversell" the validity of this assumption, but rather just explain your work as an important contribution towards the overall picture, which hopefully will include distinct distributions at some later time. The biggest weakness of the paper in my view is that the generalization from item- to user-level privacy in the context of distribution estimation is not well motivated. For instance, I understand that, for technical reasons, the authors presume all users draw their items from the same distribution, rather than distinct distributions. However, I do not understand under what scenarios such a restriction might be reasonable. Going further, I'm not sure how to assess whether user-level privacy is a useful lens for distribution estimation, since no applications are clearly linked to this constraint. It would be extremely useful for the authors to try to make explicit connections to a scenario that motivates their setting.

Correctness: To the best of my understanding, the claims in the paper are correct. There is no empirical component.

Clarity: The paper is quite well-written, and very clearly motivates the problem and walks through the intuition of the proof at the right level given the space constraints.

Relation to Prior Work: The paper is very clearly placed within the context of existing work.

Reproducibility: Yes

Additional Feedback:

[Author Response · NeurIPS 2020]

We thank all reviewers for their valuable comments and feedback. Please see replies to comments below.

## Reviewer 1

**High probability bounds:** We agree with the reviewer that obtaining a bound with dependence on the confidence
parameter is a very interesting problem. Our algorithm guarantees (upper bounds) can be extended to high probability
bounds with a $\log(1/\beta)$ dependence, where $\beta$ is the error probability. We provided guarantees in expectation as it
makes expressing min-max rates and obtaining lower bounds easier. If accepted, we will add a discussion regarding the
high probability bounds in the final version.

## Reviewer 2

We thank the reviewer for the comments. We will highlight the specific novelties such as restricted estimators and
bounds on total variation distance between binomial distributions in the final version.

## Reviewer 3

We thank the reviewer for suggestions on the writing including adding pseudo-code for the algorithms. If accepted, we
will incorporate them in the final version.

**I.i.d. data:** Distributed learning of discrete distributions when samples are generated from a single same distribution
has been studied extensively with traditional item-level differential privacy including Duchi et al. (2013), Kairouz et al.
(2016) (local differential privacy), and Diakonikolas et al. (2015), Acharya et al. (2020) (global differential privacy). It
is also common in communication constrained settings such as Barnes et al. (2019).

Our work extends these results into user-level privacy and is the first step towards understanding utility-privacy trade-offs
in user-level privacy. We believe such an analysis would give insights to design algorithms that might perform well in
the non i.i.d. setting. We agree that extending the results to distinct distributions is an interesting future direction.

**Different number of samples per user:** We have proposed a modified algorithm for the case when users have different
number of examples (see lines 141-143). Due to space constraints, the details are described in Appendix E. The user
complexity is similar to the case when all users have $m$ samples, with $m$ replaced by the median of number of user
samples. We will highlight this in the main paper.

**Finer instance-specific bounds and other metrics:** We agree that these problems are interesting and would explore
them in future works.

**Nodal differential privacy:** We thank the reviewer for the reference. We will add a discussion on it and explore future
work in this direction.

**Lemma 3:** We are inverting the function $y = (1 - p)^m$ to compute the $p$. Such an estimator is only statistically
efficient when $p$ is small. This is quantified by an upper bound on $p$, given by $c/m$. Hence, we apply this subroutine for
$p < c/m$, where $c$ can be as small as 2 or 3.

## Reviewer 4

We thank the reviewer for the positive comments.

**I.i.d. data:** Distributed learning of discrete distributions when samples are generated from a single distribution has
been studied extensively with traditional item-level differential privacy extensively including Duchi et al. (2013),
Diakonikolas et al. (2015), Kairouz et al. (2016), and Acharya et al. (2020). Our work extends these results into
user-level privacy and is the first step towards understanding utility-privacy tradeoffs in user-level privacy. We agree
that extending the results to distinct distributions is an interesting future research direction. We finally note that learning
discrete distributions with user-level DP is an important practical problem for applications such as word prediction in
virtual mobile keyboards (with/without federated learning).

• Duchi et al. (2013): Local privacy and statistical minimax rates.
• Diakonikolas et al. (2015): Differentially private learning of structured discrete distributions
• Kairouz et al. (2016): Discrete distribution estimation under local privacy
• Acharya et al. (2020): Differentially private assouad, fano, and le cam.
• Barnes et al. (2019): Lower bounds for learning distributions under communication constraints via fisher
information.

[Meta-Review · NeurIPS 2020]

This paper examines user-level privacy in the context of learning discrete distributions. Near matching upper and lower bounds (with a corresponding mechanism) on the number of users m required for a desired level of total variation distance are established, while it is shown that natural baselines the Laplace and Gaussian mechanism achieve inferior performance by a factor of sqrt(m). User-level privacy is a variation on pure/approximate differential privacy in which a mechanism's response distribution must be indistinguishable not only to change of an individual item (record) but those items belonging to a user. The paper considers the cases of users contributing equal and unequal numbers of items. R3 highlights an important concern with the user-privacy definition required for the proposed mechanism: that the users' items are drawn i.i.d. from the same distribution. This meta reviewer agrees that this assumption limits the immediate practical grounding of the work somewhat, however one can imagine situations where this assumption is reasonable - such as hospitals (users) each sharing large numbers of items drawn from one district's population. Moreoever as the reviewers note, the presented results are an important milestone in the study of user complexity and the cost of privacy in this federated learning setting; the lower bound of Theorem 3 applies to the stronger model. As R3 requests, it is paramount that the authors discuss the i.i.d. assumption in the paper. The mechanism itself is an interested application of Bun et al. (2019)'s private hypothesis selection algorithm. Along the way to achieving its goals, the paper contributes results of independent interest such as its lower bounding technique (with comparison to Acharaya et al. 2018 needed) and Theorem 5 for learning binomial distributions (bounding parameter estimation from binomial density estimation). It would be interesting to see high-probability bounds on accuracy compared to the TV-distance (expected \ell_1) and the dependence on confidence.